# Xihe: Scalable Zero-shot Time Series Learner via Hierarchical Interleaved Block Attention

## Abstract

The rapid advancement of time series foundation models (TSFMs) has been propelled by migrating architectures from language models. While existing TSFMs demonstrate impressive performance, their direct adoption of cross-domain architectures constrains effective capture of multiscale temporal dependencies inherent to time series data. This limitation becomes particularly pronounced during zero-shot transfer across datasets with divergent underlying patterns and sampling strategies. To address these challenges, we propose Hierarchical Interleaved Block Attention (HIBA) which employs hierarchical inter- and intra-block sparse attention to effectively capture multi-scale dependencies. The Intra-block attention facilitates localized contextual information exchange within individual blocks, while the inter-block attention operates across blocks to capture global temporal pattern interactions and the dynamic evolution of patterns. Leveraging the HIBA architecture, we introduce Xihe, a scalable TSFM family spanning from an ultra-efficient 9.5M parameter configuration to high-capacity 1.5B variant. Evaluated on the comprehensive GIFT-Eval benchmark, our most compact Xihe-tiny model (9.5M) surpasses the majority of contemporary TSFMs, demonstrating remarkable parameter efficiency. More impressively, Xihe-max (1.5B) establishes new state-of-the-art zero-shot performance, surpassing previous best results as of September 2025. This consistent performance excellence across the entire parameter spectrum provides compelling evidence for the exceptional generalization capabilities and architectural superiority of Xihe.

## 1 Introduction

Time series forecasting constitutes a fundamental component of decision-making and scientific analysis (Young & Shellswell, 1972; Zhang et al., 2023) across diverse domains. Time series data, while widespread across domains, is frequently scarce in individual contexts, motivating ongoing efforts to develop forecasting methods with strong cross-domain and zero-shot transfer capabilities (Oreshkin et al., 2019). Inspired by the remarkable success of foundation models in NLP, time series foundation models(TSFMs) have emerged rapidly (Ansari et al., 2024a; Das et al., 2023; Cohen et al., 2024; Liu et al., 2025; Woo et al., 2024a; Auer et al., 2025; Darlow et al., 2024). These methods leverage large-scale pre-training on multi-source datasets comprising hundreds of billions of data points to achieve impressive zero-shot forecasting performance that exceeds conventional approaches.

TSFMs have benefited from both the migration of successful transformer based design principles from language models (Ansari et al., 2024b; Das et al., 2023; Cohen et al., 2024; Liu et al., 2024; 2025; Woo et al., 2024b; Darlow et al., 2024) and the development of architecture innovations unique to time series data (Ekambaram et al., 2024; Auer et al., 2025; Graf et al., 2025). Despite notable progress, current transformer architecture based TSFMs remain constrained by architectural legacies inherited from natural language processing (NLP). One of the fundamental differences between language and time series lies in scale. In NLP,

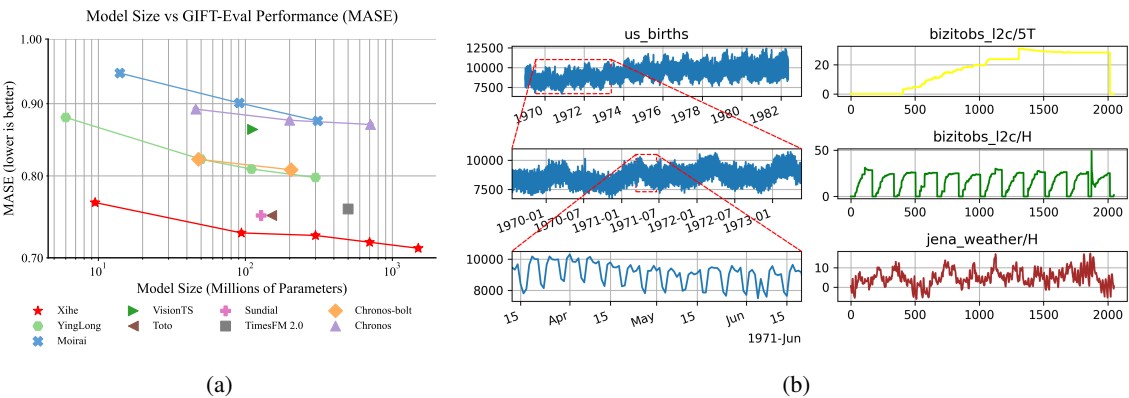

(a)            (b)

Figure 1: (a): The GIFT-Eval performance and parameter sizes of Xihe and existing TSFMs. Xihe achieves comparable, if not better, performance with less parameters. (b): Multi-scale dependencies in time series are prevalent and exhibit domain-specific characteristics. Effectively capturing these dependencies is essential for TSFMs to achieve optimal zero-shot performance. *Left*: The us_birth data is shown at different scales from top to bottom, highlighting the global trend, annual patterns, and local weekly patterns. *Right*: The scale of temporal dependencies differ as the domain and sampling strategies change across different series.

well-trained tokenizers and embedding layers learns representations for local semantics which can transfer across different linguistic contexts and domains (Cotterell et al., 2018; Chalkidis et al., 2020; Hwang et al., 2025). Attention mechanisms, in turn, are particularly effective at modeling long-range dependencies among tokens. However, as shown in Figure 1b, time series exhibit intricate multi-scale characteristics. Depending on the domain, intrinsic characteristics of the time series (e.g., seasonality, trend), and sampling strategies, the temporal spans of local dependencies (e.g., short-term patterns, short cycles) and global dependencies (e.g., long-term trends, long seasonality) can vary substantially across scales. Aiming at zero-shot transferability across different time series domains, effectively capturing both local and global dependencies across scales is therefore essential, yet remains a fundamental challenge for building a TSFM. Existing Transformer-based TSFMs, which rely on point-wise or patch-wise tokenization with the standard Transformer architecture, have failed to address this challenge.

To address these challenges, we propose a novel Hierarchical Interleaved Block Attention (HIBA) mechanism. HIBA hierarchically partitions a sequence into blocks of varying granularity and alternates intra- and inter-block attention to capture multiscale local and global dependencies. To enhance model generalization, we construct a data-quality weighted pre-training corpus by combining publicly available datasets with synthetically generated data. Leveraging the HIBA architecture, we present Xihe[1], a scalable TSFM family spanning from an ultra-efficient 9.5M parameter configuration to high-capacity 1.5B variant. Zero-shot performance of Xihe on GIFT-Eval follows a clear scaling trend, with the most compact Xihe-tiny surpasses the majority of contemporary TSFMs, demonstrating remarkable parameter efficiency. More impressively, the largest Xihe-max establishes new state-of-the-art zero-shot performance while remaining relatively efficient, as shown in Figure 1a. Our contributions are summarized as follows:

- We propose a novel attention mechanism HIBA that hierarchically partitions time series into blocks of varying sizes and alternates intra- and inter-block attention, enabling effective modeling of multi-scale long- and short-term dependencies across diverse domains and sampling frequencies.

---

[1]Xihe is a solar goddess in Chinese mythology who drives the sun in a chariot each day. Her story evokes cyclic, ordered patterns of time—much like time series track recurring temporal dynamics.

- Based on HIBA, we introduce Xihe, a family of TSFMs ranging from 9.5M to 1.5B parameters, trained on a 325B time points data corpus, with samples weighted by data quality and enriched via augmentation and synthetic generation.

- Xihe exhibit clear scaling laws in our extensive empirical evaluation. The Xihe-tiny (9.5M) and Xihe-lite (94M) achieve a well-balanced trade-off between forecasting accuracy and inference efficiency, surpassing the performance of most zero-shot models while delivering high inference throughput. The largest Xihe-max (1.5B) model demonstrates state-of-the-art zero-shot performance on the GIFT-Eval benchmark, while remaining efficient and suitable for practical deployment.

## 2 RELATED WORK

### 2.1 TIME SERIES FOUNDATION MODELS

The large-scale pre-training paradigm successfully applied in NLP has inspired time series domain moving towards universal large TSFMs which have strong zero-shot ability and effectively address data-scarce scenarios. Early works attempt to directly utilize the sequence modeling ability of large language models (LLM) (Nate Gruver & Wilson, 2023) or extend existing LLMs to adapt to time series domain (Jin et al., 2024; Sun et al., 2024). With the advancement of research, increasing efforts have been devoted to large-scale pretraining aiming to build TSFMs on massive time series corpus. Studies like Chronos, TimesFM, Moirai and Sundial (Ansari et al., 2024b; Das et al., 2024; Woo et al., 2024a; Liu et al., 2025) directly adopt the classical Transformer encoder–decoder or decoder-only architectures. Moirai-MoE (Liu et al., 2024) and Time-MoE (Shi et al., 2024) utilize mixture-of-expert (MoE) structure to achieve a better balance between model capacity and efficiency. The above methods directly borrow the model architectures of foundation models from LLMs and computer vision, which are not well-suited for capturing the unique characteristics of time series data. TTM (Ekambaram et al., 2024) utilizes a lightweight architecture composed of Multi-Layer Perceptrons (MLP). Although it achieves promising results, this architecture is not easily scalable to larger models, which limits its zero-shot performance. In contrast, our proposed model Xihe is based on HIBA mechanism, which is designed to better adapt to the diverse characteristics of time series data while maintaining the scalability of standard Transformer architecture.

### 2.2 MULTI-SCALE TIME SERIES MODELING

Multi-temporal resolution has consistently been a fundamental component in shaping the design of time series models. Early approaches typically processed each time point independently, adopting a point-scale modeling paradigm (Bai et al., 2018; Zhou et al., 2021; Salinas et al., 2020). PatchTST (Nie et al., 2022) introduces a patch-scale modeling scheme, where the time series are divided into equal-sized segments (patches) for further modeling. Many subsequent works, including some TSFMs, adopted this patch-scale strategy, which helps to suppress high-frequency noise and better model local dependencies in time series. In contrast, iTransformer and some MLP-based methods like N-BEATS and DLinear (Liu et al., 2023; Oreshkin et al., 2019; Zeng et al., 2022), take a series-scale view for time-series modeling and utilize fully-connected layers to map the whole series to hidden representations. These methods are more computationally efficient and capture global dependencies in time series more effectively. Nevertheless, all the above approaches take a single-scale view when modeling time series, thus failing to capture the complex local/global dependencies comprehensively. N-Hits and Pyraformer (Challu et al., 2023; Liu et al., 2022) perform multi-scale modeling of time series data in a hierarchical manner, but they have not explored pre-training time series foundation models on large-scale datasets with strong zero-shot capabilities; Although Moirai (Woo et al., 2024b) employs different patch sizes for series with varying sampling frequencies, it still restricts each series to a single-scale view, and its predefined mapping between frequency and patch size reduces generalization.

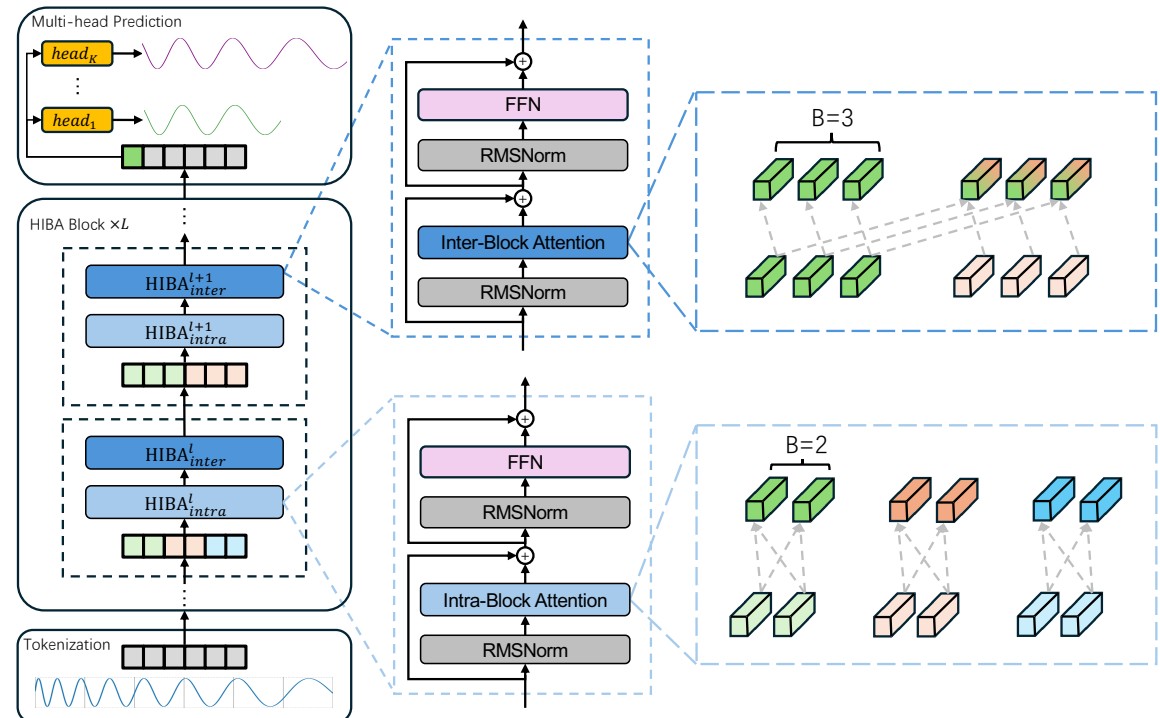

Figure 2: The Xihe architecture for time series forecasting. The time series are first patched and tokenized to embedding, then processed by our HIBA module. A multi-head prediction module is utilized to produce final forecasting. The core of our method is the Multi-Scale Attention Module, detailed on the right, which hierarchically captures temporal patterns. It comprises two components: Inter-block attention models long-range, global dependencies by performing attention across entire blocks of tokens; Intra-block attention captures local patterns by applying self-attention only within each token block.

To the best of our knowledge, Xihe is the first transformer based TSFM with multi-scale modeling, which allows it to better capture temporal dependencies at different scales and transfer more effectively in zero-shot settings across diverse time series datasets.

## 3 XIHE

In this work we focus on the time series forecasting task, which can be formally expressed as: Given historical observations of $T$ time steps $x_{1:T} = (x_1, ..., x_T) \in \mathbb{R}^T$, the objective is to learn a mapping function $\mathbb{R}^T \longrightarrow \mathbb{R}^H$ that predicts future H-step values $x_{T+1:T+H} = f_\theta(x_{1:T}) \in \mathbb{R}^H$.

The overall architecture of our Xihe model is shown in Figure 2, which consists of three components. Firstly, a tokenizer is utilized to convert original time series data $x_{1:T}$ to sequences of fine-grained hidden representations $\mathbf{h}_{1:n}$ for further modeling. Secondly, the hidden representations are processed by $L$ Hierarchical Interleaved Block Attention (HIBA) blocks to extract multi-scale temporal dependencies, denoted as

$$\mathbf{h}_{1:n}^0 = \mathbf{h}_{1:n}, \tag{1}$$

$$\mathbf{h}_{1:n}^{l+1} = \text{HIBA}_{\text{block}}^l(\mathbf{h}_{1:n}^l), \tag{2}$$

where $\text{HIBA}^l_{\text{block}}$ is the $l$-th HIBA block and $\mathbf{h}^l_{1:n}$ is the hidden representation after the $l$-th block. Finally, a multi-head prediction module produces final predictions on different quantile levels across multiple forecasting horizons. The detailed design of these components is presented in the following sections.

## 3.1 TOKENIZATION

Following Nie et al. (2022), we adopt a patch-based tokenization strategy. Before tokenization, each raw time series $x_{1:T}$ is preprocessed with InstanceNorm to produce a standard input $x'_{1:T}$ for further patching and representation extraction, formulated as:

$$x'_{1:T} = \frac{x_{1:T} - \mu_x}{\sigma_x},\tag{3}$$

where $\mu_x$ and $\sigma_x$ are the mean and standard deviation of $x_{1:T}$. We then segment the normalized series $x'_{1:T}$ into non-overlapping patches $\mathbf{x}_{1:n}$ with patch size $P$, such that $\mathbf{x}_i = x_{1+(i-1)P:iP} \in \mathbb{R}^P, i \in \{1, 2, ..., n = \lceil T/P \rceil\}$. Note that if the original sequence length is not divisible by $P$, we apply left-padding with zeros to ensure an integer number of $n$ patches. We select a relatively small patch size (8) in Xihe compared to other TSFMs that also use patch tokenization (Woo et al., 2024b), as we would like to get a more fine-grained representation to make the most of our following HIBA structure. We use a binary mask $\mathbf{m}_i \in \{0, 1\}^P$ with the same size as $\mathbf{x}_i$ to indicate the padded value or the missing value. It is then concatenate with the patched sequence to be further processed by the input embedding layer as

$$\mathbf{h}_i = \text{InputEmbed}(\text{Concat}(\mathbf{x}_i, \mathbf{m}_i)),\tag{4}$$

where $\mathbf{h}_i \in \mathbb{R}^d$ is the token embedding of $i$-th token, $d$ is the size of hidden dimension. InputEmbed is a two-layer Multi-layer Perceptron (MLP) with SiLU as activation function (Elfwing et al., 2018).

## 3.2 HIERARCHICAL INTERLEAVED BLOCK ATTENTION (HIBA)

As we mentioned in Sec. 1, most existing transformer-based TSFMs rely on token embedding for local information modeling and attention mechanism for global dependencies capturing. However, pretrained with fixed token size, these foundation models are not able to adapt to diverse time series data with drastically different temporal resolutions, seasonality, trend and sampling strategies. To overcome these limitations, we introduce HIBA, which hierarchically divide the hidden representations into different sized blocks and iteratively conduct intra- and inter-block attention to model multi-scale dependencies. While TSMixer (Ekambaram et al., 2023) and TTM (Ekambaram et al., 2024) pioneer hierarchical architectures through fixed MLP projection, their patch-mixing paradigm discards segment temporal order—impeding multi-scale temporal dependency capture for non-stationary series and degrading zero-shot generalization. Our HIBA preserve sequential granularity with temporal dependency and dynamically modeling co-evolving scales through query-key-value interactions over context-enriched blocks. The detailed description is presented as follows. For clarity of presentation, we omit the superscript $l$ whenever it is not essential.

Before processed by $\text{HIBA}_{\text{block}}$, hidden representations $h_{1:n}$ are first divided into $M$ equal sized blocks, denoted as

$$\mathbf{h}_{b,m} = \mathbf{h}_{(m-1)\times B+b}, b \in \{1, 2, ...B\}, \quad m \in \{1, ..., M = N/B\},\tag{5}$$

where $B$ is the block size, $M$ is the number of blocks. Equation 5 describes two equivalent subscript notation for the hidden state representation $\mathbf{h}$. $b$ denotes $b$-th token within divided blocks and m denotes $m$-th divided blocks. Next, two HIBA layers are employed to model the blocked $\mathbf{h}$. Both layers share a similar structure with a standard Transformer layer: they use RMSNorm as the normalization layer, a GLU with SiLU activation as the feed-forward network (FFN), and incorporate two residual connections across Attention and FFN layers. However, unlike the fully connected attention operation in standard Transformers, these two HIBA layers (denoted as $\text{HIBA}_{\text{intra}}$ and $\text{HIBA}_{\text{inter}}$) employ intra-block and inter-block attention,

respectively. In intra-block attention, a non-causal multi-head self-attention ($\text{MSA}^{\text{non-causal}}$) is applied to the hidden representations within each block, enabling thorough information fusion inside the block to capture local dependencies in time series; in inter-block attention, the representations of different blocks are processed by a causal multi-head self-attention ($\text{MSA}^{\text{causal}}$), which enables information exchange across blocks and captures global dependencies in time series while keeping causality. The whole HIBA block can be formulated as:

$$\mathbf{h}_{b,\cdot}^{\text{intra}} = \text{RMSNorm}(\mathbf{h}_{b,\cdot} + \text{MSA}^{\text{non-causal}}(\mathbf{h}_{b,\cdot})), \tag{6}$$

$$\mathbf{h}^{\text{intra\_ff}} = \text{RMSNorm}(\mathbf{h}^{\text{intra}} + \text{FFN}(\mathbf{h}^{\text{intra}}), \tag{7}$$

$$\mathbf{h}_{\cdot,m}^{\text{inter}} = \text{RMSNorm}(\mathbf{h}_{\cdot,m}^{\text{inter\_ff}} + \text{MSA}^{\text{causal}}(\mathbf{h}_{\cdot,m}^{\text{inter\_ff}})), \tag{8}$$

$$\mathbf{h}^{\text{inter\_ff}} = \text{RMSNorm}(\mathbf{h}^{\text{inter}} + \text{FFN}(\mathbf{h}^{\text{inter}})) . \tag{9}$$

By assigning different block sizes $B$ to different HIBA blocks, intra- and inter-block attention can capture local and global information at multiple scales, thereby enhancing the zero-shot transferability of the model across diverse time series datasets. The code implementation of HIBA is provided in Algorithm 1 of Appendix A.

### 3.3 MULTI-HEAD PREDICTION AND QUANTILE LOSS

Our prediction module consists of $K$ prediction heads, where each head $\text{head}_k$ corresponds to a specific horizon $H_k$ ($H_1 < H_2 < \cdots < H_K$). For the representation of each patch in the final hidden representations $h_{1:n}^L$ after $L$ HIBA blocks, $\text{head}_k$ would produce the quantile prediction of the next $H_k$ time points as:

$$\hat{x}_{i \times P + 1 : i \times P + H_k}^q = \text{head}_k(\mathbf{h}_i, q), \tag{10}$$

where $q \in Q = \{0.1, 0.2, ...0.9\}$ is the predefined quantile level. The multi-head prediction design offers several advantages. First, the temporal dependencies to be modeled often differ substantially across prediction horizons, and multiple heads encourage the model to capture the full range of information more effectively. Second, compared to the autoregressive schemes adopted by many existing TSFMs (Liu et al., 2024; Ansari et al., 2024b; Das et al., 2024) for long-horizon forecasting, using direct longer-horizon heads avoids error accumulation and does not compromise performance on short-horizon predictions. The quantile loss for $\text{head}_k$ is presented below as:

$$\mathcal{L}_k = \frac{1}{N H_k |Q|} \sum_{i=1}^{N} \sum_{t=1}^{H_k} \sum_{q \in Q} \begin{cases} q(x_{i \times P + t} - \hat{x}_{i \times P + t}^q), & \text{if } \hat{x}_{i \times P + t}^q \leq x_{i \times P + t}, \\ (1 - q)(\hat{x}_{i \times P + t}^q - x_{i \times P + t}), & \text{else.} \end{cases} \tag{11}$$

And the final loss function is the sum of losses on all prediction heads

$$\mathcal{L} = \frac{1}{K} \sum_{k=1}^{K} \mathcal{L}_k . \tag{12}$$

Note that, since non-causal attention is applied in intra-block attention, some predictions from $\mathbf{h}_i^L$ may involve acausal dependency propagation. Specifically, the non-causal intra-block attention can learn acausal patterns that, in turn, influence representations processed by the subsequent causal inter-block attention. During inference, the model operates strictly causally and does not access future horizon information. We regard these as auxiliary tasks to enhance information exchange and fusion. As there are always predictions without leakage (e.g., the last patch $\mathbf{h}_N^L$, which makes Xihe remains leakage-free at inference) the model is still able to retain robust predictive capability. An ablation study of the causality of intra-block attention is provided in Sec. 4.4.

## 4 Experiments

### 4.1 Experimental Settings

**Pretraining Datasets.** Our pretraining datasets, totaling 325 billion time points, consist of three components: (1) the LOTSA datasets from Moirai (Woo et al., 2024a), (2) subsets of the training datasets from Chronos (Ansari et al., 2024a), and (3) synthetic time series generated using a procedure inspired by KernelSynth in (Ansari et al., 2024a). Also, we utilize the Amplitude Modulation and Censor Augmentation method proposed in Auer et al. (2025) to augment the corpus during training and further increase the diversity of our data. These heterogeneous time series in the pretraining data span a wide range of sampling frequencies, diverse domains, and varying sequence lengths, enabling the training of a flexible zero-shot forecasting model. Motivated by the importance of data quality and data mixing in large language model training (Dubey et al., 2024), we adopt a data-quality–aware mixing strategy instead of the uniform mixing commonly used in prior TSFMs (Ansari et al., 2024b; Das et al., 2024). Specifically, we categorize each dataset into different levels of predictability based on its periodicity, trend strength, and noise level. During training, datasets with higher predictability are sampled with higher probability. More details for synthetic data generation and data mixing are presented in Appendix E.

**Evaluation Benchmarks.** We adopt the public time-series forecasting leaderboard GIFT-Eval benchmark(Aksu et al., 2024) (Data details in Appendix B), which comprises 23 datasets containing over 144,000 time series, spanning seven domains and ten sampling frequencies, with multivariate inputs and prediction horizons ranging from short- to long-term forecasts. The diversity of datasets and evaluation settings enables a comprehensive assessment of a model's forecasting capabilities across varied scenarios. Our pretraining datasets have no overlap with the GIFT-Eval benchmark, and the Xihe models are evaluated in a fully zero-shot setting across 97 evaluation configurations. Performance is measured using two metrics: the Mean Absolute Scaled Error (MASE) for point forecasts, and the Continuous Ranked Probability Score (CRPS) for probabilistic forecasts. To ensure comparability across benchmarks, both metrics are normalized against the Seasonal Naive baseline, and the geometric mean is then computed across all evaluation settings.

**Baseline Models.** We compare Xihe with a broad set of state-of-the-art models, including zero shot transformer based TSFMs and task-specific models. Transformer based TSFMs include Moirai (Woo et al., 2024a), Chronos/Chronos bolt(Ansari et al., 2024a), TimesFM(Das et al., 2023), Sundial(Liu et al., 2025), Toto(Cohen et al., 2024), Yinglong(Wang et al., 2025), TimeMOE(Shi et al., 2024) and VisionTS(Chen et al., 2024). Task specific models include models such as DeepAR(Flunkert et al., 2017), DLinear(Zeng et al., 2022), PatchTST(Nie et al., 2022), TFT(Lim et al., 2019), N-BEATS(Oreshkin et al., 2019) and iTransformer(Liu et al., 2023) which fits dataset-level in-distribution data. The comparison between Xihe and other Transformer-based TSFMs demonstrates HIBA approach's competitive performance relative to models employing standard attention mechanisms.

**Xihe Family.** We have developed five models for Xihe family: **Xihe-max** with 1.5 billion parameters, **Xihe-base** with 700 million parameters, **Xihe-flash** with 300 million parameters, **Xihe-lite** with 94 million parameters, **Xihe-tiny** with 9.5 million parameters (Further details in Appendix A).

### 4.2 Zero-shot Forecasting

The overall performances of Xihe on GIFT-Eval benchmark is shown on the left side of Figure 3 (see full results in Appendix C). We can tell that Xihe series achieves top zero-shot performance, **Xihe-max**, **Xihe-base** and **Xihe-flash** outperform all compared models across aggregation results; **Xihe-tiny** and **Xihe-lite** achieves comparable performance with much smaller model size. Compared with the second best zero-shot model Toto base, **Xihe-lite** demonstrates significantly superior performance with 1.7% and 2.8% reduction in CRPS and MASE respectively, while requiring fewer parameters; Compared with Moirai2, which is

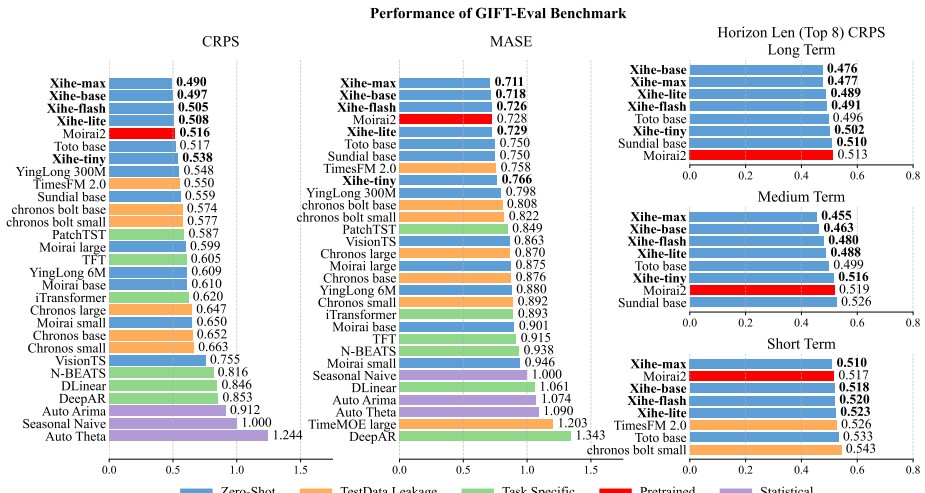

Figure 3: Results for GIFT-Eval benchmark. Aggregated probabilistic metrics CRPS (Left Panel) and point metrics MASE (Middle Panel) scores (Lower is better) of the overall benchmark and short-, medium- and long-term CRPS (Right Panel) performances (Top 8). "TestData Leakage" denotes models that have been partially trained on the benchmark datasets. "Pretrained" indicates that the benchmark training datasets were included in the model's training corpus, but without direct data leakage from the test set. "Zero-Shot" refers to models whose pre-training data contained neither the benchmark training set nor the test set.

utilize the training set in GIFT-Eval data, **Xihe-lite** obtains generally comparable results. All these results demonstrate the strong zero-shot generalization capability of our HIBA structure.

The rightmost part of Figure 3 demonstrate the aggregated metric across diverse prediction length from short to long term in GIFT-Eval benchmark measures model's ability to capture short- and long- term forecasting pattern. Xihe family show competitive performance in all forecasting horizon length compared with others models, which shows the effectiveness of HIBA and multi-head prediction module.

We further compare the inference throughput of the Xihe family with other zero-shot models under identical hardware configurations (1 x NVIDIA A100-80G GPU). As shown in Figure 4a, **Xihe-lite** and **Xihe-tiny** achieve exceptionally high throughput together with outstanding inference efficiency. Moreover, according to Figure 3, **Xihe-lite** also demonstrates superior predictive performance compared to other zero-shot models. These results suggest that the Xihe family with HIBA architecture offers a promising direction for improving inference efficiency while maintaining strong forecasting accuracy in zero-shot time-series forecasting tasks, highlighting its potential for development and deployment of time-series foundation models in resource-constrained environments.

### 4.3 SCALABILITY

Scaling laws are crucial for the development of TSFMs as they provide a principled framework for predicting expected performance gains and enable research community to allocate efforts more effectively toward key architecture designs. Figure 4b illustrates the relationship between model size and zero-shot performance of Xihe on the GIFT-Eval leaderboard. As the model size increases, both CRPS and MASE scores decrease monotonically, indicating consistent performance improvements. These results confirm that HIBA architecture within Xihe family preserves the scaling behavior observed in standard Transformers for time-series forecasting (Yao et al., 2024), and can effectively scale beyond 1B parameters.

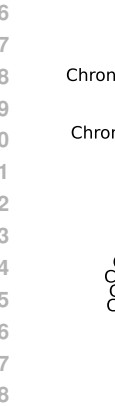 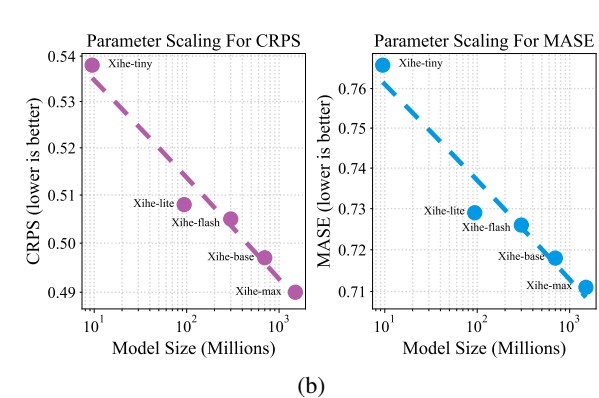

Throughput of Zero-shot Models

(a) (b)

Figure 4: (a) Throughput comparison between Xihe family and other zero-shot models, where higher values indicate greater efficiency. For each sample, the look-back window length is set to the maximum supported by the compared models, and the prediction horizon is fixed at 720. (b) Zero-shot scaling characteristics of Xihe across different model sizes on the GIFT-Eval benchmark. The left panel illustrates the scaled CRPS as a function of model size, while the right panel presents the scaled MASE against model size. Each panel includes five data points corresponding to checkpoints ranging from 9.5M to 1.5B parameters.

## 4.4 ABLATION STUDY

Table 1: Ablation studies. (**Left**) Overall MASE and CRPS scores of GIFT-Eval benchmark across different model backbone components. "Standard attn" denotes that backbone adopts the standard attention architecture. "HIBA$_{intra}$ Causal attn" indicates that the HIBA$_{intra}$ block employs causal multi-head self-attention. (**Right**) Analysis of various model prediction heads with different output patch configurations.

| | MASE | CRPS |
|---|---|---|
| **Xihe-base** | **0.718** | **0.497** |
| w/ Standard attn | 0.736 | 0.507 |
| w/ (B=3) | 0.729 | 0.505 |
| w/ (B=7) | 0.727 | 0.503 |
| w/ (B=(21, 7, 3)) | 0.719 | **0.497** |
| w/ (B=(3, 7, 21, 42)) | 0.720 | 0.505 |
| w/ HIBA$_{intra}$ Causal attn | 0.721 | 0.502 |

| | MASE | CRPS |
|---|---|---|
| **Xihe-base** | 0.718 | **0.497** |
| w/ output {96} | 0.748 | 0.537 |
| w/ output {768} | 0.720 | 0.502 |
| w/ output {96, 480, 768} | **0.717** | 0.498 |
| w/ output {96, 480, 600, 768} | 0.718 | **0.497** |

To validate the HIBA design of Xihe models, we conducted a detailed ablation study on key architectural components across the GIFT-Eval benchmark. Core results are shown in Table 1. More details ablations is presented in Appendix D.

**HIBA Ablations.** We conduct ablations on the design choices of HIBA, the results are shown in the left part of Table 1. First, We replace the HIBA in **Xihe-base** with vanilla attention and perform the model training and evaluation under identical settings. Compared with HIBA, overall MASE and CRPS increase from 0.718/0.497 to 0.736/0.537 separately, highlights the performance boost provided by HIBA. Second, we replace the non-causal multi-head attention with causal attention within each the HIBA$_{intra}$ block, causing MASE and CRPS increase from 0.718/0.497 to 0.721/0.502, implying the necessity of local information fusion with non-causal attention. Third, instead of using hierarchical block sizes in HIBA, we adopts uniform

block size 3 and block size 7 for every block, which leads to a performance drop. This shows that the hierarchical design of HIBA helps to better model multi-scale information in time series. The hierarchical block sizes setting for Xihe family is (3,7,21). Using too many block sizes, such as (3, 7, 21, 42), can also degrade performance. Under a fixed total depth, introducing an excessive number of block sizes reduces the effective number of feature-extraction cycles within stacked layers, which in turn may diminish the model's representational capacity. Furthermore, reversing order hierarchical block sizes (21,7,3) archive comparable performance with Xihe-base which shows that the model's performance is relatively insensitive to the ordering of block sizes.

**Prediction Heads Ablations.** The output horizons for multiple prediction heads in the Xihe family is $\{96, 768\}$. As shown in the right side of Table 1, **Xihe-base** with multiple prediction heads outperforms single-head design ($\{96\}$ or $\{768\}$). This indicates that joint training across multiple horizons encourages the model to learn complex temporal dependencies that generalize across forecast lengths. The results also show that adding too many prediction heads does not yield further performance gains, suggesting that combination of long prediction head and short prediction head is sufficient to maintain strong predictive performance.

The above observed ablation results is consistent across model sizes, ablation for Xihe-tiny is presented in D.4.

## 5 CONCLUSION

In this paper, we introduce Xihe, a family of time series foundation models which offers great transfer ability across time series data with multi-scale temporal dependencies. The key innovation of Xihe is the Hierarchical Interleaved Block Attention (HIBA) structure which is designed to better capture the multi-scale local and global information with intra- and inter-block attentions. Our comprehensive experiments exhibits the impressive zero-shot forecasting capability of the Xihe model, surpassing existing approaches in both accuracy and efficiency. In the future, we would expand Xihe to larger sizes to further push the limit of TSFMs. Also, Xihe still limited to uni-variate time series forecasting, the framework could be adapted to multivariate inputs via shared embeddings per time step or channel-specific encodings. And we leave the extension to multivariate forecasting task, additional tasks (e.g., classification and anomaly detection) and the incorporation of richer information (e.g., covariates or multi-domain information) as future work.

## ETHICS STATEMENT

The authors have adhered to the ICLR Code of Ethics. This work is a technical contribution of time series forecasting using publicly available datasets (e.g., traffic, weather) which, to our knowledge, do not contain personally identifiable information. This work does not present foreseeable direct ethical harms. We urge practitioners applying our method to consider the potential for amplifying data-driven biases and to assess the societal impact of their specific use case.

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

## A  IMPLEMENTATION DETAILS

All experiments are implemented using Pytorch and performed with NVIDIA A100 GPUs. We use the Adam optimizer for model optimization and cosine scheduler for learning rate scheduler type. The initial learning rate is 0.0001 and the warmup ratio is set to be 0.01. During training, all training samples are mixed according to a specific ratio to ensure that the model can learn temporal patterns across diverse domains. Model configurations of Xihe family in different sizes are provided in Table 2.

Table 2: Model configurations of the Xihe family. $d$ is the embedding dimension of Transformer. $d_{ff}$ is the hidden dimension of FFN. $(H_q, H_{kv}$ denotes number of query heads and number of key/value heads separately.

| Model | Patch Size | Context Length | Prediction Length | Layers | Dimension $(d, d_{ff})$ | MHA Heads $(H_q, H_{kv})$ | HIBA Block size $B$ | Total Parameters #Count |
|-------|-----------|----------------|-------------------|--------|-----------|-----------|-----------|------------------|
| Xihe-tiny | 8 | 2688 | $\{96, 768\}$ | 24 | (160, 640) | (10, 2) | (3,7,21) | **9.5M** |
| Xihe-lite | 8 | 2688 | $\{96, 768\}$ | 24 | (448, 2432) | (14,2) | (3,7,21) | **94M** |
| Xihe-flash | 8 | 2688 | $\{96, 768\}$ | 24 | (896, 4864) | (14,2) | (3,7,21) | **300M** |
| Xihe-base | 8 | 2688 | $\{96, 768\}$ | 48 | (896, 4864) | (14,2) | (3,7,21) | **700M** |
| Xihe-max | 8 | 2688 | $\{96, 768\}$ | 96 | (896, 4864) | (14,2) | (3,7,21) | **1.5B** |

---

**Algorithm 1** Pseudocode for HIBA Model Configurations of Xihe-base

---

1: **Input:** Input time series $x \in \mathbb{R}^{2688}$, binary mask $m \in \mathbb{R}^{2688}$, Patch Size $p = 8$, Context Length $ctx = 2688$, Layers $L = 48$, Dimension $d = 896$, Dimension $d_{ff} = 4864$, query heads $H_q = 14$, key/value heads $H_{kv} = 2$ and HIBA Block size $B = (3, 7, 21)$.

2: $\mathbf{x} \leftarrow \text{PATCH}(\mathbf{x})$        $\triangleright \mathbf{x} \in \mathbb{R}^{336 \times 8}$

3: $\mathbf{m} \leftarrow \text{PACH}(\mathbf{m})$        $\triangleright \mathbf{m} \in \mathbb{R}^{336 \times 8}$

4: $\mathbf{h} \leftarrow \text{CONCAT}(\mathbf{x}, \mathbf{m})$        $\triangleright \mathbf{h} \in \mathbb{R}^{336 \times 16}$

5: $\mathbf{h} \leftarrow \text{INPUTEMBED}(\mathbf{h})$        $\triangleright \mathbf{h} \in \mathbb{R}^{336 \times d}$

6: **function** HIBA_INTRA($h, b$)

7:      $\mathbf{h} \leftarrow \text{RESHAPE}(\mathbf{h})$        $\triangleright \mathbf{h} \in \mathbb{R}^{(336/b) \times b \times d}$

8:      $\mathbf{h}_{attn} \leftarrow \text{MSA}(\mathbf{h}, H_q, H_{kv})$        $\triangleright \mathbf{h}_{attn} \in \mathbb{R}^{(336/b) \times b \times d}$

9:      $\mathbf{h} \leftarrow \text{RMSNORM}(\mathbf{h} + \mathbf{h}_{attn})$        $\triangleright \mathbf{h} \in \mathbb{R}^{(336/b) \times b \times d}$

10:      $\mathbf{h} \leftarrow \text{RESHAPE}(\mathbf{h})$        $\triangleright \mathbf{h} \in \mathbb{R}^{336 \times d}$

11:      $\mathbf{h}_{ffn} \leftarrow \text{FFN}(\mathbf{h}, d_{ff})$        $\triangleright \mathbf{h}_{ffn} \in \mathbb{R}^{336 \times d}$

12:      $\mathbf{h} \leftarrow \text{RMSNORM}(\mathbf{h} + \mathbf{h}_{ffn})$        $\triangleright \mathbf{h} \in \mathbb{R}^{336 \times d}$

13:      **return h**

14: **end function**

15: **function** HIBA_INTER($h, b$)

16:      $\mathbf{h} \leftarrow \text{RESHAPE}(\mathbf{h})$        $\triangleright \mathbf{h} \in \mathbb{R}^{(336/b) \times b \times d}$

17:      $\mathbf{h} \leftarrow \text{TRANSPOSE}(\mathbf{h})$        $\triangleright \mathbf{h} \in \mathbb{R}^{b \times (336/b) \times d}$

18:      $\mathbf{h}_{attn} \leftarrow \text{MSA}(\mathbf{h}, H_q, H_{kv})$        $\triangleright \mathbf{h}_{attn} \in \mathbb{R}^{b \times (336/b) \times d}$

19:      $\mathbf{h} \leftarrow \text{RMSNORM}(\mathbf{h} + \mathbf{h}_{attn})$        $\triangleright \mathbf{h} \in \mathbb{R}^{b \times (336/b) \times d}$

20:      $\mathbf{h} \leftarrow \text{TRANSPOSE}(\mathbf{h})$        $\triangleright \mathbf{h} \in \mathbb{R}^{(336/b) \times b \times d}$

21:      $\mathbf{h} \leftarrow \text{RESHAPE}(\mathbf{h})$        $\triangleright \mathbf{h} \in \mathbb{R}^{336 \times d}$

22:      $\mathbf{h}_{ffn} \leftarrow \text{FFN}(\mathbf{h}, d_{ff})$        $\triangleright \mathbf{h}_{ffn} \in \mathbb{R}^{336 \times d}$

23:      $\mathbf{h} \leftarrow \text{RMSNORM}(\mathbf{h} + \mathbf{h}_{ffn})$        $\triangleright \mathbf{h} \in \mathbb{R}^{336 \times d}$

24:      **return h**

25: **end function**

26: $c \leftarrow \text{COUNT}(B)$        $\triangleright$ count the number of elements in $B$,in this case $c = 3$

27: **for** $\ell = 1$ to $(L // (c * 2))$ **do**

28:      **for** $b$ in $B$ **do**

29:          $\mathbf{h} \leftarrow \text{HIBA\_INTRA}(\mathbf{h}, b)$

30:          $\mathbf{h} \leftarrow \text{HIBA\_INTER}(\mathbf{h}, b)$

31:      **end for**

32: **end for**

---

# B GIFT-EVAL BENCHMARK

Table 3: Individual statistics of GIFT-Eval benchmark across all datasets.

| Dataset | Source | Domain | Frequency | # Series | Series Length | | | # Obs |
| | | | | | Avg | Min | Max | |
|---|---|---|---|---|---|---|---|---|
| Jena Weather | Autoformer (Wu et al., 2021) | Nature | 10T | 1 | 52,704 | 52,704 | 52,704 | 52,704 |
| Jena Weather | Autoformer (Wu et al., 2021) | Nature | H | 1 | 8,784 | 8,784 | 8,784 | 8,784 |
| Jena Weather | Autoformer (Wu et al., 2021) | Nature | D | 1 | 366 | 366 | 366 | 366 |
| BizITObs - Application | AutoMixer (Palaskar et al., 2024) | Web/CloudOps | 10S | 1 | 8,834 | 8,834 | 8,834 | 8,834 |
| BizITObs - Service | AutoMixer (Palaskar et al., 2024) | Web/CloudOps | 10S | 21 | 8,835 | 8,835 | 8,835 | 185,535 |
| BizITObs - L2C | AutoMixer (Palaskar et al., 2024) | Web/CloudOps | 5T | 1 | 31,968 | 31,968 | 31,968 | 31,968 |
| BizITObs - L2C | AutoMixer (Palaskar et al., 2024) | Web/CloudOps | H | 1 | 2,664 | 2,664 | 2,664 | 2,664 |
| Bitbrains - Fast Storage | Grid Workloads Archive (Shen et al., 2015) | Web/CloudOps | 5T | 1,250 | 8,640 | 8,640 | 8,640 | 10,800,000 |
| Bitbrains - Fast Storage | Grid Workloads Archive (Shen et al., 2015) | Web/CloudOps | H | 1,250 | 721 | 721 | 721 | 901,250 |
| Bitbrains - rnd | Grid Workloads Archive (Shen et al., 2015) | Web/CloudOps | 5T | 500 | 8,640 | 8,640 | 8,640 | 4,320,000 |
| Bitbrains - rnd | Grid Workloads Archive (Shen et al., 2015) | Web/CloudOps | H | 500 | 720 | 720 | 720 | 360,000 |
| Restaurant | Recruit Rest. Comp. (Howard et al., 2017) | Sales | D | 807 | 358 | 67 | 478 | 289,303 |
| ETT1 | Informer (Zhou et al., 2020) | Energy | 15T | 1 | 69,680 | 69,680 | 69,680 | 69,680 |
| ETT1 | Informer (Zhou et al., 2020) | Energy | H | 1 | 17,420 | 17,420 | 17,420 | 17,420 |
| ETT1 | Informer (Zhou et al., 2020) | Energy | D | 1 | 725 | 725 | 725 | 725 |
| ETT1 | Informer (Zhou et al., 2020) | Energy | W-THU | 1 | 103 | 103 | 103 | 103 |
| ETT2 | Informer (Zhou et al., 2020) | Energy | 15T | 1 | 69,680 | 69,680 | 69,680 | 69,680 |
| ETT2 | Informer (Zhou et al., 2020) | Energy | H | 1 | 17,420 | 17,420 | 17,420 | 17,420 |
| ETT2 | Informer (Zhou et al., 2020) | Energy | D | 1 | 725 | 725 | 725 | 725 |
| ETT2 | Informer (Zhou et al., 2020) | Energy | W-THU | 1 | 103 | 103 | 103 | 103 |
| Loop Seattle | LibCity (Wang et al., 2023a) | Transport | 5T | 323 | 105,120 | 105,120 | 105,120 | 33,953,760 |
| Loop Seattle | LibCity (Wang et al., 2023a) | Transport | H | 323 | 8,760 | 8,760 | 8,760 | 2,829,480 |
| Loop Seattle | LibCity (Wang et al., 2023a) | Transport | D | 323 | 365 | 365 | 365 | 117,895 |
| SZ-Taxi | LibCity (Wang et al., 2023a) | Transport | 15T | 156 | 2,976 | 2,976 | 2,976 | 464,256 |
| SZ-Taxi | LibCity (Wang et al., 2023a) | Transport | H | 156 | 744 | 744 | 744 | 116,064 |
| M_DENSE | LibCity (Wang et al., 2023a) | Transport | H | 30 | 17,520 | 17,520 | 17,520 | 525,600 |
| M_DENSE | LibCity (Wang et al., 2023a) | Transport | D | 30 | 730 | 730 | 730 | 21,900 |
| Solar | LSTNet (Lai et al., 2017) | Energy | 10T | 137 | 52,560 | 52,560 | 52,560 | 7,200,720 |
| Solar | LSTNet (Lai et al., 2017) | Energy | H | 137 | 8,760 | 8,760 | 8,760 | 1,200,120 |
| Solar | LSTNet (Lai et al., 2017) | Energy | D | 137 | 365 | 365 | 365 | 50,005 |
| Solar | LSTNet (Lai et al., 2017) | Energy | W-FRI | 137 | 52 | 52 | 52 | 7,124 |
| Hierarchical Sales | Mancuso et al. (2020) | Sales | D | 118 | 1,825 | 1,825 | 1,825 | 215,350 |
| Hierarchical Sales | Mancuso et al. (2020) | Sales | W-WED | 118 | 260 | 260 | 260 | 30,680 |
| M4 Yearly | Monash (Godahewa et al., 2021) | Econ/Fin | A-DEC | 22,974 | 37 | 19 | 284 | 845,109 |
| M4 Quarterly | Monash (Godahewa et al., 2021) | Econ/Fin | Q-DEC | 24,000 | 100 | 24 | 874 | 2,406,108 |
| M4 Monthly | Monash (Godahewa et al., 2021) | Econ/Fin | M | 48,000 | 234 | 60 | 2,812 | 11,246,411 |
| M4 Weekly | Monash (Godahewa et al., 2021) | Econ/Fin | W-SUN | 359 | 1,035 | 93 | 2,610 | 371,579 |
| M4 Daily | Monash (Godahewa et al., 2021) | Econ/Fin | D | 4,227 | 2,371 | 107 | 9,933 | 10,023,836 |
| M4 Hourly | Monash (Godahewa et al., 2021) | Econ/Fin | H | 414 | 902 | 748 | 1,008 | 373,372 |
| Hospital | Monash (Godahewa et al., 2021) | Healthcare | M | 767 | 84 | 84 | 84 | 64,428 |
| COVID Deaths | Monash (Godahewa et al., 2021) | Healthcare | D | 266 | 212 | 212 | 212 | 56,392 |
| US Births | Monash (Godahewa et al., 2021) | Healthcare | D | 1 | 7,305 | 7,305 | 7,305 | 7,305 |
| US Births | Monash (Godahewa et al., 2021) | Healthcare | W-TUE | 1 | 1,043 | 1,043 | 1,043 | 1,043 |
| US Births | Monash (Godahewa et al., 2021) | Healthcare | M | 1 | 240 | 240 | 240 | 240 |
| Saugeen | Monash (Godahewa et al., 2021) | Nature | D | 1 | 23,741 | 23,741 | 23,741 | 23,741 |
| Saugeen | Monash (Godahewa et al., 2021) | Nature | W-THU | 1 | 3,391 | 3,391 | 3,391 | 3,391 |
| Saugeen | Monash (Godahewa et al., 2021) | Nature | M | 1 | 780 | 780 | 780 | 780 |
| Temperature Rain | Monash (Godahewa et al., 2021) | Nature | D | 32,072 | 725 | 725 | 725 | 780 |
| KDD Cup 2018 | Monash (Godahewa et al., 2021) | Nature | H | 270 | 10,898 | 9,504 | 10,920 | 2,942,364 |
| KDD Cup 2018 | Monash (Godahewa et al., 2021) | Nature | D | 270 | 455 | 396 | 455 | 122,791 |
| Car Parts | Monash (Godahewa et al., 2021) | Sales | M | 2,674 | 51 | 51 | 51 | 136,374 |
| Electricity | UCI ML Archive (Trindade, 2015) | Energy | 15T | 370 | 140,256 | 140,256 | 140,256 | 51,894,720 |
| Electricity | UCI ML Archive (Trindade, 2015) | Energy | H | 370 | 35,064 | 35,064 | 35,064 | 12,973,680 |
| Electricity | UCI ML Archive (Trindade, 2015) | Energy | D | 370 | 1,461 | 1,461 | 1,461 | 540,570 |
| Electricity | UCI ML Archive (Trindade, 2015) | Energy | W-FRI | 370 | 208 | 208 | 208 | 76,960 |

# C DETAIL BENCHMARK RESULTS

Table 4: Detailed CRPS scores of different zero-shot models on the GIFT-Eval benchmark. Lower is better. The best score is bold and the second best is underlined. At the end of table, we also count numbers of best score and second best scores.

| Dataset | Xihe-max | Xihe-lite | Toto base | Sundial base | Yinglong 300M | Moirai large |
|---|---|---|---|---|---|---|
| loop_seattle/5T/short | 0.048 | 0.049 | 0.048 | 0.05 | 0.052 | **0.041** |
| loop_seattle/5T/medium | 0.072 | 0.074 | 0.072 | 0.077 | 0.092 | **0.038** |
| loop_seattle/5T/long | 0.078 | 0.081 | 0.077 | 0.084 | 0.096 | **0.049** |
| loop_seattle/D/short | **0.04** | 0.044 | 0.044 | 0.047 | 0.043 | 0.045 |
| loop_seattle/H/short | **0.058** | 0.062 | 0.063 | 0.067 | 0.063 | 0.066 |
| loop_seattle/H/medium | **0.062** | 0.067 | 0.064 | 0.075 | 0.067 | 0.07 |
| loop_seattle/H/long | **0.061** | 0.064 | 0.065 | 0.072 | 0.068 | 0.074 |
| m_dense/D/short | **0.067** | 0.071 | 0.075 | **0.067** | 0.073 | 0.095 |
| m_dense/H/short | 0.134 | 0.138 | 0.148 | 0.133 | 0.156 | **0.128** |
| m_dense/H/medium | 0.119 | 0.119 | 0.121 | 0.128 | 0.134 | **0.112** |
| m_dense/H/long | 0.118 | 0.118 | 0.128 | 0.13 | 0.145 | **0.114** |
| sz_taxi/15T/short | 0.204 | 0.205 | **0.203** | 0.223 | **0.203** | 0.215 |
| sz_taxi/15T/medium | 0.204 | 0.205 | 0.205 | 0.228 | **0.203** | 0.215 |
| sz_taxi/15T/long | 0.199 | 0.199 | 0.202 | 0.221 | **0.198** | 0.213 |
| sz_taxi/H/short | 0.138 | 0.139 | **0.137** | 0.154 | **0.137** | 0.146 |
| bitbrains_fast_storage/5T/short | 0.418 | 0.448 | **0.371** | 0.462 | 0.424 | 0.412 |
| bitbrains_fast_storage/5T/medium | 0.647 | 0.668 | **0.629** | 0.728 | 0.645 | 0.636 |
| bitbrains_fast_storage/5T/long | 0.754 | 0.802 | **0.669** | 0.811 | 0.709 | 0.716 |
| bitbrains_fast_storage/H/short | 0.712 | 0.748 | **0.623** | 0.764 | 0.631 | 0.646 |
| bitbrains_rnd/5T/short | 0.436 | 0.448 | **0.399** | 0.433 | 0.425 | 0.418 |
| bitbrains_rnd/5T/medium | 0.623 | 0.635 | 0.628 | 0.73 | 0.652 | **0.594** |
| bitbrains_rnd/5T/long | **0.588** | 0.604 | 0.589 | 0.715 | 0.689 | 0.678 |
| bitbrains_rnd/H/short | 0.602 | 0.638 | 0.593 | 0.725 | 0.673 | **0.566** |
| bizitobs_application/10S/short | **0.009** | 0.011 | 0.012 | 0.016 | 0.017 | 0.038 |
| bizitobs_application/10S/medium | **0.019** | 0.029 | 0.034 | 0.046 | 0.048 | 0.084 |
| bizitobs_application/10S/long | 0.055 | 0.054 | **0.053** | 0.061 | 0.061 | 0.094 |
| bizitobs_l2c/5T/short | 0.076 | 0.076 | 0.069 | **0.067** | 0.077 | 0.079 |
| bizitobs_l2c/5T/medium | 0.365 | 0.386 | 0.316 | **0.234** | 0.379 | 0.41 |
| bizitobs_l2c/5T/long | 0.544 | 0.553 | 0.533 | **0.31** | 0.576 | 0.508 |
| bizitobs_l2c/H/short | 0.223 | 0.202 | **0.199** | 0.223 | 0.229 | 0.559 |
| bizitobs_l2c/H/medium | 0.25 | **0.235** | 0.356 | 0.276 | 0.33 | 0.619 |
| bizitobs_l2c/H/long | 0.28 | **0.274** | 0.369 | 0.325 | 0.406 | 0.6 |
| bizitobs_service/10S/short | **0.011** | 0.012 | **0.011** | 0.016 | 0.017 | 0.032 |
| bizitobs_service/10S/medium | **0.019** | 0.026 | 0.027 | 0.044 | 0.045 | 0.069 |
| bizitobs_service/10S/long | 0.054 | 0.053 | **0.051** | 0.057 | 0.062 | 0.104 |
| car_parts/M/short | 0.965 | 0.993 | **0.899** | 1.189 | 1.191 | 1.18 |
| covid_deaths/D/short | 0.032 | 0.037 | **0.027** | 0.131 | 0.078 | 0.046 |
| electricity/15T/short | 0.092 | 0.099 | 0.099 | **0.084** | 0.093 | 0.128 |
| electricity/15T/medium | **0.077** | 0.081 | 0.086 | 0.082 | 0.079 | 0.103 |
| electricity/15T/long | **0.076** | 0.079 | 0.086 | 0.082 | 0.078 | 0.099 |
| electricity/D/short | **0.054** | 0.056 | 0.059 | 0.064 | **0.054** | 0.069 |
| electricity/H/short | **0.041** | 0.059 | 0.069 | 0.069 | 0.078 | 0.077 |
| electricity/H/medium | **0.039** | 0.057 | 0.075 | 0.08 | 0.082 | 0.087 |
| electricity/H/long | **0.043** | 0.062 | 0.083 | 0.093 | 0.097 | 0.103 |
| electricity/W/short | **0.041** | 0.048 | 0.064 | 0.072 | 0.057 | 0.062 |
| ett1/15T/short | **0.162** | 0.165 | **0.162** | 0.177 | 0.166 | 0.226 |
| ett1/15T/medium | 0.247 | 0.249 | 0.26 | 0.26 | **0.243** | 0.342 |
| ett1/15T/long | 0.245 | 0.246 | 0.251 | 0.253 | **0.234** | 0.358 |
| ett1/D/short | 0.285 | **0.267** | 0.284 | 0.373 | 0.284 | 0.286 |
| ett1/H/short | **0.182** | 0.186 | 0.194 | 0.19 | **0.182** | 0.189 |
| ett1/H/medium | 0.253 | 0.263 | 0.254 | 0.269 | **0.252** | 0.27 |
| ett1/H/long | 0.266 | 0.269 | 0.267 | 0.283 | **0.264** | 0.296 |
| ett1/W/short | **0.25** | 0.265 | 0.263 | 0.404 | 0.27 | 0.26 |
| ett2/15T/short | 0.069 | 0.069 | 0.068 | 0.069 | **0.066** | 0.08 |
| ett2/15T/medium | 0.093 | 0.099 | 0.093 | 0.096 | **0.09** | 0.105 |
| ett2/15T/long | 0.097 | 0.095 | **0.088** | 0.098 | 0.092 | 0.115 |
| ett2/D/short | 0.094 | 0.095 | 0.111 | 0.103 | **0.092** | 0.094 |
| ett2/H/short | **0.064** | 0.065 | 0.065 | 0.072 | **0.064** | 0.069 |
| ett2/H/medium | 0.109 | **0.1** | 0.102 | 0.114 | 0.104 | 0.118 |
| ett2/H/long | 0.111 | **0.107** | 0.108 | 0.117 | **0.107** | 0.125 |
| ett2/W/short | 0.096 | **0.09** | 0.106 | 0.098 | 0.091 | 0.109 |
| hierarchical_sales/D/short | 0.583 | 0.577 | **0.57** | 0.649 | 0.589 | 0.58 |
| hierarchical_sales/W/short | **0.349** | 0.355 | 0.356 | 0.39 | 0.371 | 0.359 |
| hospital/M/short | 0.055 | 0.055 | 0.052 | 0.061 | 0.057 | **0.051** |
| jena_weather/10T/short | 0.03 | 0.029 | **0.027** | 0.031 | 0.03 | 0.051 |
| jena_weather/10T/medium | 0.052 | 0.052 | **0.049** | 0.054 | 0.051 | 0.072 |
| jena_weather/10T/long | **0.05** | **0.05** | **0.05** | 0.056 | 0.052 | 0.077 |
| jena_weather/D/short | **0.045** | 0.046 | 0.051 | 0.048 | 0.05 | 0.051 |
| jena_weather/H/short | 0.044 | 0.044 | **0.042** | 0.05 | 0.045 | 0.045 |
| jena_weather/H/medium | **0.052** | **0.052** | 0.053 | 0.058 | 0.057 | 0.058 |
| jena_weather/H/long | 0.058 | **0.057** | **0.057** | 0.066 | 0.06 | 0.061 |

**Table 4 continued from previous page**

| Dataset | Xihe-max | Xihe-lite | Toto base | Sundial base | Yinglong 300M | Moirai large |
|---|---|---|---|---|---|---|
| kdd_cup_2018/D/short | 0.39 | 0.385 | 0.387 | 0.396 | **0.374** | 0.381 |
| kdd_cup_2018/H/short | 0.381 | 0.394 | 0.403 | **0.351** | 0.374 | 0.362 |
| kdd_cup_2018/H/medium | 0.434 | 0.457 | 0.441 | **0.377** | 0.417 | 0.387 |
| kdd_cup_2018/H/long | 0.461 | 0.468 | 0.457 | **0.375** | 0.439 | 0.378 |
| m4_daily/D/short | **0.021** | 0.021 | 0.022 | 0.027 | 0.023 | 0.03 |
| m4_hourly/H/short | 0.021 | 0.021 | 0.035 | 0.023 | 0.025 | **0.02** |
| m4_monthly/M/short | **0.093** | 0.095 | 0.097 | 0.116 | 0.104 | 0.095 |
| m4_quarterly/Q/short | 0.076 | 0.077 | 0.078 | 0.093 | 0.086 | **0.073** |
| m4_weekly/W/short | **0.039** | 0.04 | 0.049 | 0.043 | 0.041 | 0.046 |
| m4_yearly/A/short | 0.116 | 0.115 | 0.122 | 0.16 | 0.152 | **0.104** |
| restaurant/D/short | **0.258** | 0.26 | 0.297 | 0.286 | 0.266 | 0.27 |
| saugeen/D/short | 0.368 | 0.371 | **0.353** | 0.379 | 0.381 | 0.406 |
| saugeen/M/short | 0.326 | 0.337 | **0.299** | 0.332 | 0.328 | 0.324 |
| saugeen/W/short | 0.381 | 0.399 | 0.39 | 0.406 | **0.36** | 0.43 |
| solar/10T/short | 0.549 | 0.611 | 0.541 | **0.444** | 0.553 | 0.596 |
| solar/10T/medium | 0.367 | 0.367 | 0.353 | 0.373 | **0.348** | 0.747 |
| solar/10T/long | **0.347** | 0.348 | 0.352 | 0.365 | 0.351 | 0.771 |
| solar/D/short | 0.288 | 0.29 | 0.29 | 0.324 | **0.278** | 0.292 |
| solar/H/short | **0.326** | 0.353 | 0.328 | 0.329 | 0.355 | 0.333 |
| solar/H/medium | 0.325 | 0.358 | 0.331 | **0.309** | 0.374 | 0.346 |
| solar/H/long | 0.338 | 0.342 | 0.331 | **0.293** | 0.352 | 0.347 |
| solar/W/short | 0.141 | **0.139** | 0.186 | 0.148 | 0.255 | 0.213 |
| temperature_rain/D/short | 0.57 | 0.569 | 0.56 | 0.62 | 0.571 | **0.479** |
| us_births/D/short | **0.02** | 0.021 | 0.026 | 0.022 | 0.026 | 0.027 |
| us_births/M/short | 0.017 | **0.013** | **0.013** | 0.028 | 0.015 | 0.016 |
| us_births/W/short | 0.015 | **0.013** | 0.014 | 0.017 | 0.015 | 0.018 |
| **rank 1** | 32 | 13 | 24 | 11 | 19 | 13 |
| **rank 2** | 29 | 33 | 23 | 1 | 11 | 12 |
| **rank sum** | 61 | 46 | 47 | 12 | 30 | 25 |

Table 5: Detailed MASE scores of different zero-shot models on the GIFT-Eval benchmark. Lower is better. The best score is bold and the second best is underlined. At the end of table, we also count numbers of best score and second best scores.

| Dataset | Xihe-max | Xihe-lite | Toto base | Sundial base | Yinglong 300M | Moirai large |
|---|---|---|---|---|---|---|
| loop_seattle/5T/short | 0.559 | 0.559 | 0.562 | 0.542 | 0.607 | **0.486** |
| loop_seattle/5T/medium | 0.802 | 0.814 | 0.804 | 0.82 | 1.023 | **0.45** |
| loop_seattle/5T/long | 0.864 | 0.887 | 0.848 | 0.893 | 1.07 | **0.556** |
| loop_seattle/D/short | **0.818** | 0.871 | 0.925 | 0.9 | 0.907 | 0.916 |
| loop_seattle/H/short | **0.823** | 0.876 | 0.899 | 0.88 | 0.895 | 0.945 |
| loop_seattle/H/medium | **0.908** | 0.974 | 0.929 | 1.014 | 0.966 | 1.0 |
| loop_seattle/H/long | **0.899** | 0.925 | 0.943 | 0.987 | 0.981 | 1.05 |
| m_dense/D/short | 0.715 | 0.747 | 0.763 | **0.681** | 0.745 | 0.957 |
| m_dense/H/short | 0.785 | 0.809 | 0.879 | 0.791 | 0.929 | **0.777** |
| m_dense/H/medium | 0.707 | 0.709 | 0.728 | 0.759 | 0.788 | **0.684** |
| m_dense/H/long | 0.72 | 0.72 | 0.78 | 0.771 | 0.843 | **0.696** |
| sz_taxi/15T/short | **0.548** | 0.551 | 0.55 | 0.554 | 0.551 | 0.581 |
| sz_taxi/15T/medium | **0.537** | 0.54 | 0.545 | 0.563 | 0.541 | 0.569 |
| sz_taxi/15T/long | 0.512 | 0.513 | 0.518 | 0.537 | **0.511** | 0.554 |
| sz_taxi/H/short | **0.563** | 0.568 | 0.568 | 0.581 | 0.568 | 0.601 |
| bitbrains_fast_storage/5T/short | 0.722 | 0.761 | **0.672** | 0.74 | 0.803 | 0.827 |
| bitbrains_fast_storage/5T/medium | 0.994 | 1.038 | **0.985** | 1.108 | 1.072 | 1.02 |
| bitbrains_fast_storage/5T/long | 0.902 | 0.938 | **0.897** | 1.011 | 1.01 | 0.955 |
| bitbrains_fast_storage/H/short | 1.084 | 1.141 | **0.945** | 1.15 | 1.116 | 1.09 |
| bitbrains_rnd/5T/short | 1.685 | 1.75 | **1.65** | 1.715 | 1.786 | 1.75 |
| bitbrains_rnd/5T/medium | **4.405** | 4.461 | 4.417 | 4.562 | 4.498 | 4.46 |
| bitbrains_rnd/5T/long | 3.345 | 3.389 | **3.337** | 3.522 | 3.47 | 3.42 |
| bitbrains_rnd/H/short | 5.846 | 5.937 | **5.638** | 5.98 | 5.892 | 5.93 |
| bizitobs_application/10S/short | **1.013** | 1.044 | 1.247 | 1.429 | 1.818 | 4.51 |
| bizitobs_application/10S/medium | **1.68** | 2.149 | 2.304 | 2.857 | 3.868 | 7.39 |
| bizitobs_application/10S/long | 3.267 | **3.186** | 3.275 | 3.705 | 4.6 | 7.84 |
| bizitobs_l2c/5T/short | 0.276 | 0.277 | 0.259 | **0.248** | 0.286 | 0.285 |
| bizitobs_l2c/5T/medium | 0.817 | 0.891 | 0.754 | **0.53** | 0.877 | 0.987 |
| bizitobs_l2c/5T/long | 1.077 | 1.134 | 1.177 | **0.635** | 1.214 | 1.12 |
| bizitobs_l2c/H/short | 0.533 | 0.486 | **0.47** | 0.476 | 0.554 | 1.15 |
| bizitobs_l2c/H/medium | 0.527 | **0.495** | 0.757 | 0.55 | 0.707 | 1.25 |
| bizitobs_l2c/H/long | 0.608 | **0.591** | 0.797 | 0.665 | 0.868 | 1.27 |
| bizitobs_service/10S/short | 0.797 | **0.767** | 0.789 | 0.839 | 1.138 | 2.31 |

Table 5 – continued from previous page

| Dataset | Xihe-max | Xihe-lite | Toto Base | Sundial base | Yinglong 300M | Moirai large |
|---|---|---|---|---|---|---|
| bizitobs_service/10S/medium | **1.02** | 1.063 | 1.083 | 1.272 | 2.024 | 3.87 |
| bizitobs_service/10S/long | 1.464 | 1.389 | **1.302** | 1.457 | 2.209 | 4.33 |
| car_parts/M/short | 0.857 | 0.874 | **0.81** | 0.957 | 1.065 | 0.903 |
| covid_deaths/D/short | 35.652 | 37.947 | **32.619** | 60.375 | 45.404 | 36.5 |
| electricity/15T/short | 1.06 | 1.128 | 1.145 | **0.895** | 1.072 | 1.71 |
| electricity/15T/medium | 0.861 | 0.894 | 0.988 | **0.854** | 0.883 | 1.29 |
| electricity/15T/long | **0.898** | 0.93 | 1.044 | 0.906 | 0.918 | 1.31 |
| electricity/D/short | 1.411 | 1.436 | 1.485 | 1.456 | **1.396** | 1.51 |
| electricity/H/short | **0.527** | 0.813 | 0.976 | 0.932 | 1.092 | 1.08 |
| electricity/H/medium | **0.504** | 0.79 | 1.103 | 0.993 | 1.139 | 1.2 |
| electricity/H/long | **0.54** | 0.842 | 1.235 | 1.075 | 1.29 | 1.36 |
| electricity/W/short | **1.236** | 1.397 | 1.79 | 1.614 | 1.599 | 1.79 |
| ett1/15T/short | **0.692** | 0.706 | 0.693 | 0.71 | 0.717 | 0.925 |
| ett1/15T/medium | 1.041 | 1.045 | 1.081 | 1.067 | **1.029** | 1.3 |
| ett1/15T/long | 1.053 | 1.052 | 1.069 | 1.088 | **1.033** | 1.4 |
| ett1/D/short | 1.766 | 1.702 | **1.659** | 1.903 | 1.728 | 1.75 |
| ett1/H/short | 0.826 | 0.837 | 0.865 | **0.829** | 0.832 | 0.855 |
| ett1/H/medium | **1.244** | 1.281 | 1.272 | 1.288 | 1.26 | 1.34 |
| ett1/H/long | 1.376 | **1.354** | 1.37 | 1.405 | 1.37 | 1.45 |
| ett1/W/short | **1.467** | 1.519 | 1.579 | 1.843 | 1.589 | 1.51 |
| ett2/15T/short | 0.805 | 0.823 | 0.786 | **0.747** | 0.753 | 1.0 |
| ett2/15T/medium | 0.934 | 0.975 | 0.923 | 0.907 | **0.888** | 1.06 |
| ett2/15T/long | 0.968 | 0.948 | **0.871** | 0.921 | 0.906 | 1.14 |
| ett2/D/short | 1.356 | 1.383 | 1.615 | 1.507 | **1.3** | 1.44 |
| ett2/H/short | **0.734** | 0.749 | 0.735 | 0.771 | 0.741 | 0.783 |
| ett2/H/medium | 1.069 | 1.024 | **1.017** | 1.115 | 1.018 | 1.18 |
| ett2/H/long | 1.124 | 1.1 | 1.077 | 1.139 | **1.057** | 1.28 |
| ett2/W/short | 0.971 | **0.915** | 0.987 | 0.936 | 0.92 | 1.31 |
| hierarchical_sales/D/short | 0.746 | 0.743 | **0.735** | 0.79 | 0.763 | 0.745 |
| hierarchical_sales/W/short | **0.72** | 0.729 | 0.744 | 0.751 | 0.747 | 0.749 |
| hospital/M/short | 0.78 | 0.78 | 0.783 | 0.837 | 0.793 | **0.768** |
| jena_weather/10T/short | 0.288 | 0.28 | **0.266** | 0.297 | 0.319 | 0.338 |
| jena_weather/10T/medium | 0.62 | 0.615 | **0.598** | 0.639 | 0.617 | 0.694 |
| jena_weather/10T/long | **0.628** | 0.636 | 0.635 | 0.679 | 0.639 | 0.792 |
| jena_weather/D/short | 1.015 | 1.022 | 1.196 | **0.931** | 1.122 | 1.14 |
| jena_weather/H/short | **0.517** | 0.517 | 0.544 | 0.539 | 0.543 | 0.585 |
| jena_weather/H/medium | 0.809 | 0.803 | **0.753** | 0.869 | 0.886 | 0.891 |
| jena_weather/H/long | 0.931 | 0.951 | 1.014 | 1.088 | 1.03 | **0.881** |
| kdd_cup_2018/D/short | 1.224 | 1.201 | 1.212 | **1.174** | 1.183 | 1.2 |
| kdd_cup_2018/H/short | 0.946 | 0.963 | 0.99 | **0.801** | 0.927 | 0.894 |
| kdd_cup_2018/H/medium | 1.074 | 1.105 | 1.078 | **0.841** | 1.034 | 0.954 |
| kdd_cup_2018/H/long | 1.048 | 1.048 | 1.041 | **0.775** | 1.004 | 0.867 |
| m4_daily/D/short | **3.281** | 3.308 | 3.312 | 3.715 | 3.513 | 4.18 |
| m4_hourly/H/short | **0.774** | 0.844 | 0.861 | 0.869 | 0.925 | 0.886 |
| m4_monthly/M/short | **0.93** | 0.958 | 0.983 | 1.099 | 1.048 | 0.977 |
| m4_quarterly/Q/short | **1.194** | 1.225 | 1.227 | 1.457 | 1.39 | 1.14 |
| m4_weekly/W/short | 2.151 | **2.133** | 2.4 | 2.396 | 2.248 | 2.58 |
| m4_yearly/A/short | 3.328 | 3.295 | 3.397 | 4.33 | 4.312 | **2.97** |
| restaurant/D/short | **0.68** | 0.687 | 0.783 | 0.704 | 0.703 | 0.715 |
| saugeen/D/short | 3.038 | 2.973 | 2.965 | **2.783** | 3.135 | 3.29 |
| saugeen/M/short | 0.756 | 0.783 | 0.757 | **0.753** | 0.785 | 0.756 |
| saugeen/W/short | 1.218 | 1.238 | 1.307 | 1.198 | **1.186** | 1.38 |
| solar/10T/short | 1.007 | 1.096 | 1.033 | **0.837** | 1.107 | 1.11 |
| solar/10T/medium | 0.879 | **0.859** | 0.881 | 0.941 | 0.89 | 1.82 |
| solar/10T/long | 0.863 | **0.826** | 0.88 | 0.949 | 0.886 | 1.95 |
| solar/D/short | **0.967** | 0.981 | 1.012 | 1.081 | 0.972 | 0.987 |
| solar/H/short | 0.836 | 0.886 | 0.828 | **0.787** | 0.911 | 0.875 |
| solar/H/medium | 0.859 | 0.932 | 0.865 | **0.767** | 0.967 | 0.917 |
| solar/H/long | 0.963 | 0.946 | 0.957 | **0.749** | 0.961 | 1.02 |
| solar/W/short | **0.965** | 0.983 | 1.427 | 0.981 | 1.799 | 1.53 |
| temperature_rain/D/short | 1.371 | 1.369 | 1.365 | 1.43 | 1.417 | **1.2** |
| us_births/D/short | **0.378** | 0.407 | 0.498 | 0.389 | 0.506 | 0.503 |
| us_births/M/short | 0.856 | 0.595 | **0.581** | 1.158 | 0.73 | 0.771 |
| us_births/W/short | 1.263 | **1.067** | 1.235 | 1.362 | 1.237 | 1.47 |
| **rank 1** | 31 | 11 | 19 | 18 | 8 | 11 |
| **rank 2** | 34 | 29 | 17 | 9 | 9 | 5 |
| **rank sum** | 65 | 40 | 36 | 27 | 17 | 16 |

# D ABLATION RESULTS

## D.1 STANDARD ATTENTION ABLATION

The ablation for HIBA architecture across diverse sampling frequency is summarized in Table 6). HIBA outperform vanilla attention at majority of sampling frequencies, which indicates that the hierarchical multi-scale design provided by HIBA$_{intra}$ and HIBA$_{inter}$ provides enhanced temporal pattern characterization at varying sampling frequency and zero-shot forecasting generalization capabilities for heterogeneous time series.

Table 6: Ablation studies. MASE and CRPS scores of GIFT-Eval benchmark across different sampling frequency. "Standard attn" denotes that backbone adopts the standard attention architecture.

| MASE | Yearly | Quarterly | Monthly | Weekly | Daily | Hourly | Minutely | Secondly |
|---|---|---|---|---|---|---|---|---|
| Xihe-base | **0.838** | **0.745** | **0.852** | **0.744** | **0.676** | **0.665** | **0.756** | 0.759 |
| w/ Standard attn | 0.907 | 0.824 | 0.905 | 0.755 | 0.697 | 0.673 | 0.785 | **0.748** |

| CRPS | Yearly | Quarterly | Monthly | Weekly | Daily | Hourly | Minutely | Secondly |
|---|---|---|---|---|---|---|---|---|
| Xihe-base | **0.844** | **0.777** | **0.789** | **0.594** | **0.429** | **0.423** | **0.529** | 0.541 |
| w/ Standard attn | 0.902 | 0.835 | 0.844 | 0.607 | 0.449 | 0.422 | 0.553 | **0.496** |

## D.2 HIBA ABLATION

Table 7: Ablation studies for Xihe-base. We re-trained and evaluated each ablation five times with different random seeds. Overall average MASE and CRPS scores with confidence interval of GIFT-Eval benchmark across different model backbone components. "Standard attn" denotes that backbone adopts the standard attention architecture. "HIBA$_{intra}$ Causal attn" indicates that the HIBA$_{intra}$ block employs causal multi-head self-attention.

| | MASE | CRPS |
|---|---|---|
| **Xihe-base** | **0.718**$\pm$**0.0003** | **0.497**$\pm$**0.0005** |
| w/ Standard attn | 0.736$\pm$0.0005 | 0.507$\pm$0.0003 |
| w/ (B=3) | 0.729$\pm$0.0006 | 0.505$\pm$0.0005 |
| w/ (B=7) | 0.727$\pm$0.0004 | 0.503$\pm$0.0005 |
| w/ (B=$(21, 7, 3)$) | 0.719$\pm$0.0004 | **0.497**$\pm$**0.0004** |
| w/ (B=$(3, 7, 21, 42)$) | 0.720$\pm$0.0005 | 0.505$\pm$0.0005 |
| w/ (B=$(4, 8, 16)$) | 0.725$\pm$0.0004 | 0.506$\pm$0.0006 |
| w/ HIBA$_{intra}$ Causal attn | 0.721$\pm$0.0003 | 0.502$\pm$0.0004 |

## D.3 PREDICTION HEADS ABLATION

Suppose the prediction heads have output patch $\{96, 480, 768\}$. If the forecasting horizon of task is less than or equal to 96 steps, the model uses the prediction head with output patch 96; if the forecasting horizon of task is greater than 96 steps but less than or equal to 480 steps ,it uses the head with output patch 480;if the forecasting horizon of task exceeds 480, it uses the head with output patch 768.

Table 8: Ablation studies for Xihe-base. We re-trained and evaluated each ablation five times with different random seeds. Analysis of various model prediction heads with different output patch configurations.

|  | MASE | CRPS |
|---|---|---|
| **Xihe-base** | 0.718±0.0003 | **0.497±0.0005** |
| w/ output $\{96\}$ | 0.748±0.0005 | 0.537±0.0005 |
| w/ output $\{768\}$ | 0.720±0.0004 | 0.502±0.0006 |
| w/ output $\{96, 480, 768\}$ | **0.717±0.0004** | 0.498±0.0004 |
| w/ output $\{96, 480, 600, 768\}$ | 0.718±0.0003 | **0.497±0.0004** |

## D.4 XIHE-TINY ABLATION

Table 9: Ablation studies for Xihe-tiny. We re-trained and evaluated each ablation five times with different random seeds. Overall average MASE and CRPS scores with confidence interval of GIFT-Eval benchmark across different model backbone components. "Standard attn" denotes that backbone adopts the standard attention architecture. "HIBA$_{intra}$ Causal attn" indicates that the HIBA$_{intra}$ block employs causal multi-head self-attention.

|  | MASE | CRPS |
|---|---|---|
| **Xihe-tiny** | **0.766±0.0005** | **0.538±0.0006** |
| w/ Standard attn | 0.776±0.0004 | 0.543±0.0003 |
| w/ (B=3) | 0.772±0.0004 | 0.540±0.0005 |
| w/ (B=7) | 0.769±0.0004 | 0.539±0.0005 |
| w/ (B=$(21, 7, 3)$) | **0.766±0.0006** | **0.538±0.0004** |
| w/ (B=$(3, 7, 21, 42)$) | 0.767±0.0007 | 0.541±0.0005 |
| w/ (B=$(4, 8, 16)$) | 0.769±0.0004 | 0.542±0.0005 |
| w/ HIBA$_{intra}$ Causal attn | 0.771±0.0003 | 0.540±0.0003 |

Table 10: Ablation studies for Xihe-tiny. We re-trained and evaluated each ablation five times with different random seeds. Analysis of various model prediction heads with different output patch configurations.

|  | MASE | CRPS |
|---|---|---|
| **Xihe-tiny** | 0.766±0.0005 | **0.538±0.0006** |
| w/ output $\{96\}$ | 0.748±0.0005 | 0.537±0.0007 |
| w/ output $\{768\}$ | 0.720±0.0005 | 0.502±0.0005 |
| w/ output $\{96, 480, 768\}$ | 0.766±0.0007 | **0.538±0.0004** |
| w/ output $\{96, 480, 600, 768\}$ | **0.765±0.0004** | 0.539±0.0004 |

## D.5 DATA-QUALITY-AWARE MIXING STRATEGY

We conduct ablation studies with Xihe-base to quantify the respective contributions of data quality improvement and model architecture innovation to overall performance gains. First, we fix the model architecture and compare two data mixing strategies: (1) uniform mixing strategy and (2) data-quality-aware mixing strategy. This allows us to isolate and measure the performance gain attributable to the data mixing strategy. Then, we fix the data mixing strategy and compare different model architectures to quantify the contribution

of architectural innovations. The experimental results are summarized in the table below and have been added to the appendix. They show that, under a fixed architecture, data-quality-aware mixing strategy leads to an average reduction of 0.0105/0.0045 in MASE/CRPS. In contrast, under a fixed data mixing strategy, architectural innovations yield an average reduction of 0.0125/0.0055 in MASE/CRPS. These quantitative results indicate that the performance improvement brought by model architecture innovation is substantially larger than that achieved by data-quality-aware mixing strategy.

Table 11: Ablation studies for Data-quality-aware Mixing Strategy. (**Left**) Overall MASE scores of GIFT-Eval benchmark across different model backbone and data mixing strategy. "Standard attn" denotes that backbone adopts the standard attention architecture. "uniform" denote uniform mixing strategy and "Data-quality-aware" denote data-quality-aware mixing strategy. (**Right**) Overall CRPS scores of GIFT-Eval benchmark across different model backbone and data mixing strategy.

|  | uniform | data-quality-aware |  |  | uniform | data-quality-aware |
|---|---|---|---|---|---|---|
| **Xihe-tiny** | 0.774 | **0.766** |  | **Xihe-tiny** | 0.542 | **0.538** |
| Standard attn | 0.789 | 0.776 |  | Standard attn | 0.548 | 0.543 |

## D.6 HIBA CONFIG

In following experiments, we found that keeping 24 layers but shrinking hidden dimensions and feed-forward sizes yielded better accuracy.

Table 12: Ablation studies for HIBA config for Xihe-tiny. $d$ is the embedding dimension of Transformer. $d_{ff}$ is the hidden dimension of FFN. ($H_q, H_{kv}$ denotes number of query heads and number of key/value heads separately. Compared with Xihe-tiny-wide, Xihe-tiny shrink hidden dimension and feed-forward sizes, but have more stack layers.

| Model | MASE | CRPS | Layers | Dimension $(d, d_{ff})$ | MHA Heads $(H_q, H_{kv})$ | HIBA Block size $B$ | Total Parameters #Count |
|---|---|---|---|---|---|---|---|
| **Xihe-tiny** | **0.766** | **0.538** | 24 | (160, 640) | (10, 2) | (3,7,21) | **9.5M** |
| **Xihe-tiny-wide** | 0.771 | 0.541 | 12 | (256, 768) | (8,2) | (3,7,21) | **9.9M** |

## E PRETRAINING DATASET

In this section, we provide a detailed description of the data sources and building process of our pretraining dataset.

### E.1 REAL-WORLD DATA

Our real-world datasets consist of collections from Chronos and the LOTSA dataset, comprising over 300 billion data points. Note that, to accurately evaluate our model's zero-shot performance on the GIFT-Eval benchmark, we removed all datasets appearing in GIFT-Eval from the final training corpus to avoid data leakage. All datasets used in our final training corpus are listed in Table 13.

## E.2 SYNTHETIC DATA

We extended the KernelSynth algorithm proposed in (Ansari et al., 2024b) from three aspects to generate our synthetic data. First, we increase the maximum generated sequence length from 1024 to 4096 to match the look-back window and forecasting horizon of Xihe models. Second, We add ExpSinSquared kernels with small length scale to generate spike signals. Third, compared with the original strategy of randomly sampling multiple kernels, we impose a constraint that the sampled kernels must include either a periodic component (ExpSineSquared) or a smooth trend component (DotProduct or RBF), ensuring that the generated sequences remain forecastable. We also make modifications to the periods of sampled ExpSinSquared kernels. These extensions make the data generated by KernelSynth more closely resemble real-world time-series characteristics, even under greater diversity. The details of our Extended KernelSynth algorithm are presented in Algorithm 2.

## E.3 DATA MIXTURE

In this section, we provide a detailed description of our data-quality-aware data mixing procedure.

We first assess the data quality of each real-world dataset. Specifically, we sample multiple sequences from each dataset and compute indicators such as ACF, trend/seasonal strength after STL decomposition, Hurst exponent Haslett & Raftery (1989), and spectral entropy, which characterize periodicity, trend, and noise levels. Then, a human expert would consider these indicators together with visual inspection of the sampled sequences and assess the *forecastability* (classified as high or low) and *noise level* (classified as high, medium, or low) of each dataset. We rely on human assessment because, given the great diversity of time-series patterns across rich real-world data, no single statistical metric can adequately capture overall forecastability. We subsequently divide all real-world datasets into five groups according to the assessed labels forecastability-noise. The groups are: high–low, high–medium, high–high, low–low, and a combined low–medium/high category, where the medium and high noise levels are merged due to their similar characteristics under low forecastability. During Training, these five groups are assigned sampling probabilities of 40%, 20%, 10%, 7%, and 3%, respectively. The synthetic data is assigned sampling probability of 20%, as our extended KernelSynth algorithm guarantees a certain level of forecastability. Within each group, all datasets are sampled with equal probability to ensure that differences in dataset size do not introduce sampling imbalance. This data mixure strategy ensures that the model is trained predominantly on high-quality data while still maintaining strong generalization ability.

Table 13: Real-world datasets in training corpus

| Dataset | Domain | Frequency | # Time Series | # Time points |
|---------|--------|-----------|---------------|---------------|
| Wind Power | Energy | 4S | 1 | 7,397,147 |
| Residential Load Power | Energy | T | 813 | 437,983,677 |
| Residential PV Power | Energy | T | 699 | 376,016,850 |
| Los-Loop | Transport | 5T | 207 | 7,094,304 |
| PEMS03 | Transport | 5T | 358 | 9,382,464 |
| PEMS04 | Transport | 5T | 921 | 15,649,632 |
| PEMS07 | Transport | 5T | 883 | 24,921,792 |
| PEMS08 | Transport | 5T | 510 | 9,106,560 |
| PEMS Bay | Transport | 5T | 325 | 16,941,600 |
| Alibaba Cluster Trace 2018 | CloudOps | 5T | 116,818 | 190,385,060 |
| Azure VM Traces 2017 | CloudOps | 5T | 159,472 | 885,522,908 |
| Borg Cluster Data 2011 | CloudOps | 5T | 286,772 | 1,075,105,708 |
| LargeST | Transport | 5T | 42,333 | 4,452,510,528 |
| KDD Cup 2022 | Energy | 10T | 134 | 4,727,519 |

Table 13: Real-world datasets in training corpus

| Dataset | Domain | Frequency | # Time Series | # Time points |
|---|---|---|---|---|
| HZMetro | Transport | 15T | 160 | 380,320 |
| Q-Traffic | Transport | 15T | 45,148 | 264,386,688 |
| SHMetro | Transport | 15T | 576 | 5,073,984 |
| Beijing Subway | Transport | 30T | 552 | 867,744 |
| Elecdemand | Energy | 30T | 1 | 17,520 |
| Australian Electricity Demand | Energy | 30T | 5 | 1,155,264 |
| London Smart Meters | Energy | 30T | 5,560 | 166,528,896 |
| Taxi | Transport | 30T | 2428 | 3,589,798 |
| BDG-2 Bear | Energy | H | 91 | 1,482,312 |
| BDG-2 Fox | Energy | H | 135 | 2,324,568 |
| BDG-2 Panther | Energy | H | 105 | 919,800 |
| BDG-2 Rat | Energy | H | 280 | 4,728,288 |
| Borealis | Energy | H | 15 | 83,269 |
| BDG-2 Bull | Energy | H | 41 | 719,304 |
| China Air Quality | Nature | H | 2,622 | 34,435,404 |
| BDG-2 Cockatoo | Energy | H | 1 | 17,544 |
| Covid19 Energy | Energy | H | 1 | 31,912 |
| ELF | Energy | H | 1 | 21,792 |
| GEF12 | Energy | H | 20 | 788,280 |
| GEF14 | Energy | H | 1 | 17,520 |
| GEF17 | Energy | H | 8 | 140,352 |
| BDG-2 Hog | Energy | H | 24 | 421,056 |
| IDEAL | Energy | H | 217 | 1,255,253 |
| Low Carbon London | Energy | H | 713 | 9,543,553 |
| Oikolab Weather | Climate | H | 8 | 800,456 |
| PDB | Energy | H | 1 | 17,520 |
| Sceaux | Energy | H | 1 | 34,223 |
| SMART | Energy | H | 5 | 95,709 |
| Spanish Energy and Weather | Energy | H | 1 | 35,064 |
| ERCOT Load | Energy | H | 8 | 1,238,976 |
| Mexico City Bikes | Transport | H | 494 | 38,687,004 |
| Electricity (Hourly) | Energy | H | 321 | 8,443,584 |
| Beijing Air Quality | Nature | H | 132 | 4,628,448 |
| Pedestrian Counts | Transport | H | 66 | 3,132,346 |
| Rideshare | Transport | H | 2,304 | 859,392 |
| Traffic | Transport | H | 862 | 15,122,928 |
| Taxi (Hourly) | Transport | H | 2,428 | 1,794,292 |
| Uber TLC (Hourly) | Transport | H | 262 | 1,138,128 |
| Wind Farms (Hourly) | Energy | H | 337 | 2,869,414 |
| Weatherbench (Hourly) | Nature | H | 225,280 | 78,992,150,528 |
| Buildings900K | Energy | H | 1,795,256 | 15,728,237,816 |
| ERA5 | Climate | H | 11,059,200 | 96,613,171,200 |
| CMIP6 | Climate | 6H | 14,327,808 | 104,592,998,400 |
| Bitcoin | Econ/Fin | D | 18 | 81,918 |
| Covid Mobility | Transport | D | 362 | 148,602 |
| Extended Web Traffic | Web | D | 145,063 | 370,926,091 |
| Favorita Sales | Sales | D | 111,840 | 139,179,538 |

Table 13: Real-world datasets in training corpus

| Dataset | Domain | Frequency | # Time Series | # Time points |
|---|---|---|---|---|
| Favorita Transactions | Sales | D | 54 | 84,408 |
| Subseasonal | Climate | D | 3,448 | 56,788,560 |
| Subseasonal Precipitation | Climate | D | 862 | 9,760,426 |
| Sunspot | Nature | D | 1 | 73,894 |
| Vehicle Trips | Transport | D | 329 | 32,512 |
| Wiki-Rolling | Web | D | 47,675 | 40,619,100 |
| Dominick | Retail | D | 100,014 | 29,652,492 |
| M5 | Sales | D | 30,490 | 47,649,940 |
| Monash Weather | Climate | D | 3,010 | 43,032,000 |
| NN5 Daily | Econ/Fin | D | 111 | 87,801 |
| Uber TLC Daily | Transport | D | 262 | 47,422 |
| Weatherbench (Daily) | Nature | D | 225,280 | 3,291,336,704 |
| Wiki Daily (100k) | Web | D | 100,000 | 274,100,000 |
| Wind Farms (Daily) | Energy | D | 337 | 119,549 |
| Exchange Rate | Finance | D | 8 | 60,704 |
| CDC Fluview ILINet | Healthcare | W | 375 | 319,515 |
| CDC Fluview WHO NREVSS | Healthcare | W | 296 | 167,040 |
| Kaggle Web Traffic Weekly | Web | W | 145,063 | 16,537,182 |
| Project Tycho | Healthcare | W | 1,258 | 1,377,707 |
| Traffic Weekly | Transport | W | 862 | 82,752 |
| Electricity (Weekly) | Energy | W | 321 | 50,076 |
| NN5 Weekly | Econ/Fin | W | 111 | 12,543 |
| Weatherbench (Weekly) | Nature | W | 225,280 | 470,159,360 |
| GoDaddy | Econ/Fin | M | 6,270 | 257,070 |
| CIF 2016 | Econ/Fin | M | 72 | 7,108 |
| FRED MD | Econ/Fin | M | 107 | 77,896 |
| M1 Monthly | Econ/Fin | M | 617 | 55,998 |
| M3 Monthly | Econ/Fin | M | 1,428 | 167,562 |
| Tourism Monthly | Econ/Fin | M | 366 | 109,280 |
| M3 Other | Econ/Fin | Q | 174 | 11,933 |
| M1 Quarterly | Econ/Fin | Q | 203 | 9,944 |
| M3 Quarterly | Econ/Fin | Q | 756 | 37,004 |
| Tourism Quarterly | Econ/Fin | Q | 427 | 42,544 |
| M1 Yearly | Econ/Fin | Y | 181 | 4,515 |
| M3 Yearly | Econ/Fin | Y | 645 | 18,319 |
| Tourism Yearly | Econ/Fin | Y | 518 | 12,757 |

---

**Algorithm 2** Extended KernelSynth

---

1: **Input:** Kernel bank $\mathcal{K}$, maximum kernels per time series $J = 5$, and length of the time series $l_{\mathrm{syn}} = 4096$.
2: **Output:** A synthetic time series $x_{1:l_{\mathrm{syn}}}$.
3: $j \sim \mathcal{U}\{1, ..., J\}$                                     $\triangleright$ sample the number of kernels
4: **repeat**
5:      $\{\kappa_1(t,t'), \ldots, \kappa_j(t,t')\} \overset{\mathrm{i.i.d.}}{\sim} \mathcal{K}$             $\triangleright$ sample $j$ kernels from the kernel bank
6: **until** at least one $\kappa_i$ is periodic (ExpSineSquared) or a smooth trend kernel (DotProduct or RBF) and the number of periodic kernels is less than three.
7: **if** at least one $\kappa_i$ is periodic **then**
8:      Sample primary period

$$p_1 \sim \mathcal{U}\{4, 7, 12, 24, 52, 60, 96, 144, 168, 288, 360, \mathrm{rand}(4, 2016)\}$$

                                                      $\triangleright$ set the base period;
9:      **if** there is a second periodic component **then**
10:         $k \sim \mathcal{U}\{7, 52, \mathrm{rand}(4, 100)\}$           $\triangleright$ sample multiplier for the second period
11:         $p_2 \leftarrow k \cdot p_1$                                   $\triangleright$ define the second period
12:      **end if**
13:      Adjust the hyperparameters of the periodic kernels (e.g., their periods and length-scales) using $p_1$ and, if present, $p_2$                      $\triangleright$ encode the sampled periodic structure
14: **end if**
15: $\kappa^*(t,t') \leftarrow \kappa_1(t,t')$
16: **for** $i \leftarrow 2$ to $j$ **do**
17:      $\star \sim \{+, \times\}$                              $\triangleright$ sample a random binary operator
18:      $\kappa^*(t,t') \leftarrow \kappa^*(t,t') \star \kappa_i(t,t')$             $\triangleright$ compose kernels
19: **end for**
20: $x_{1:l_{\mathrm{syn}}} \sim \mathcal{GP}(0, \kappa^*(t,t'))$               $\triangleright$ sample from the GP prior
21: **return** $x_{1:l_{\mathrm{syn}}}$

---

## F   FORECASTING SHOWCASES

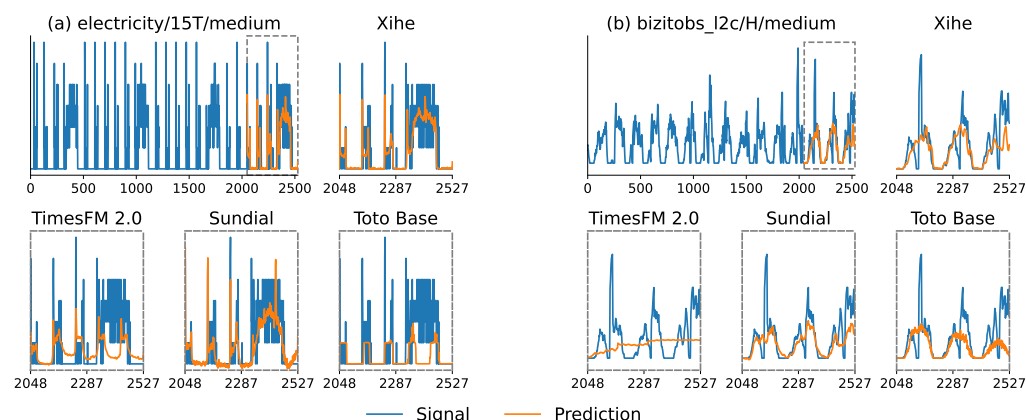

Figure 5: Two examples of forecasts comparison from GIFT-Eval benchmark. For each sample, we provide both the full context and **Xihe-max** prediction, as well as the zoomed-in prediction of other zero-shot models.

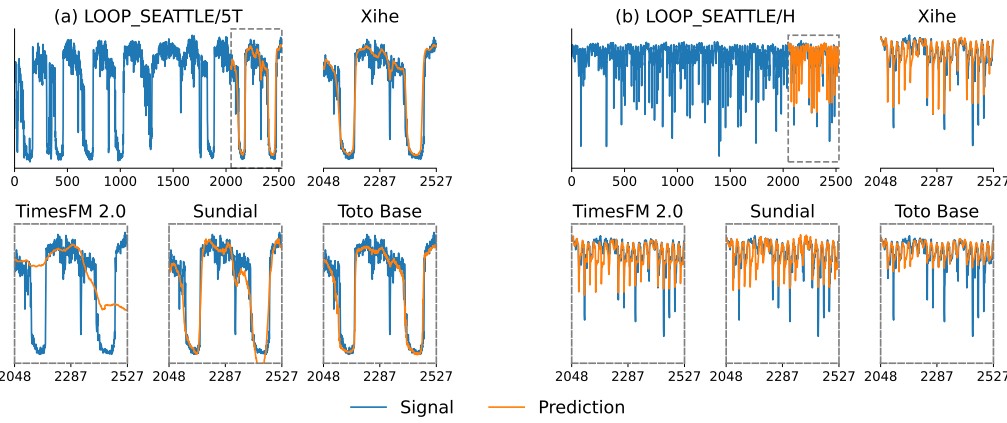

Figure 6: Examples of forecasts comparison from GIFT-Eval benchmark. For each sample, we provide both the full context and **Xihe-max** prediction, as well as the zoomed-in prediction of other zero-shot models.

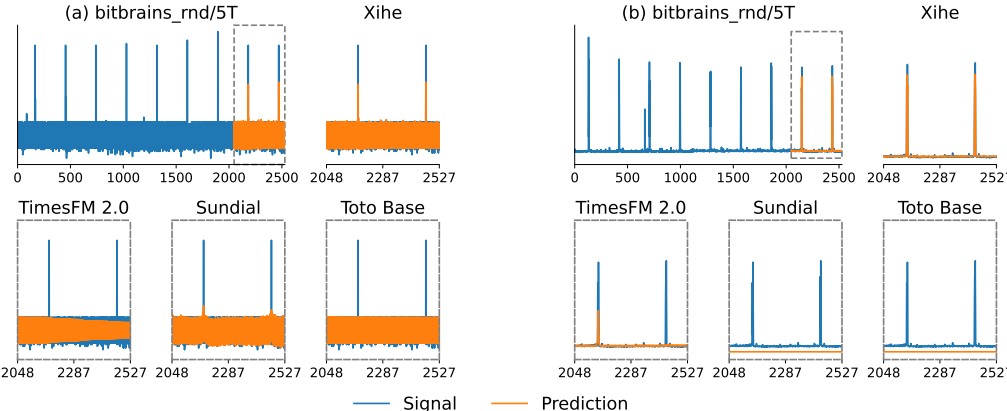

Figure 7: Examples of forecasts comparison from GIFT-Eval benchmark. For each sample, we provide both the full context and **Xihe-max** prediction, as well as the zoomed-in prediction of other zero-shot models.

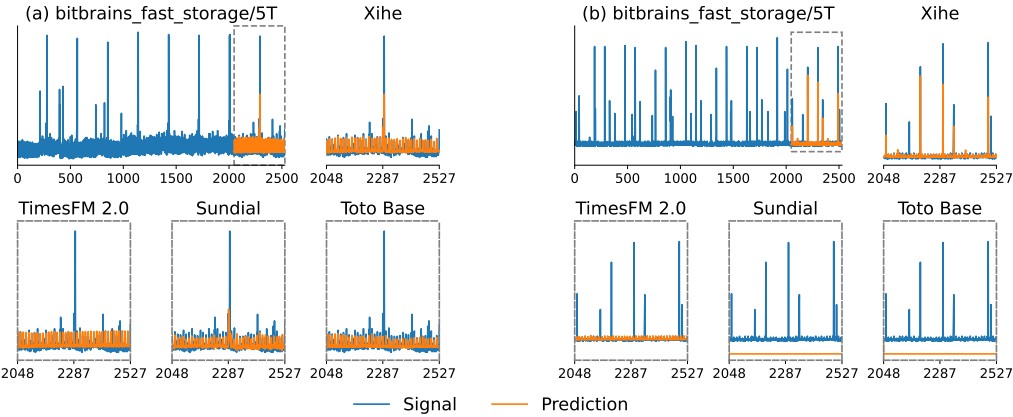

Figure 8: Examples of forecasts comparison from GIFT-Eval benchmark. For each sample, we provide both the full context and **Xihe-max** prediction, as well as the zoomed-in prediction of other zero-shot models.

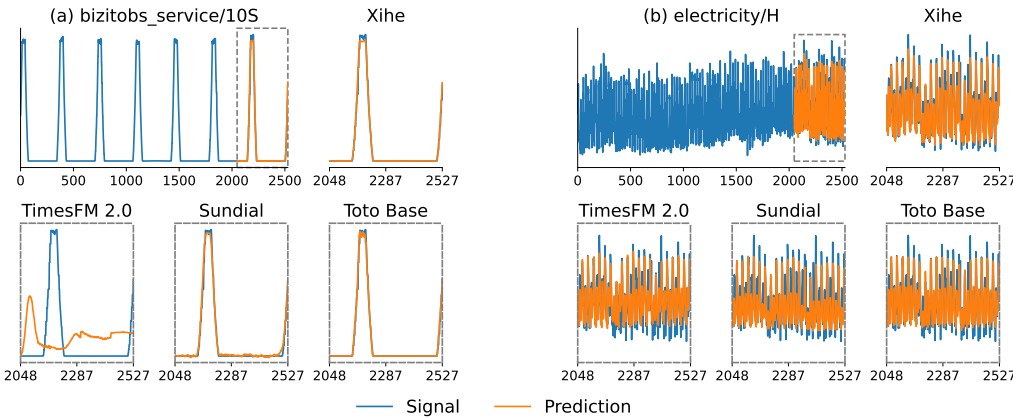

Figure 9: Examples of forecasts comparison from GIFT-Eval benchmark. For each sample, we provide both the full context and **Xihe-max** prediction, as well as the zoomed-in prediction of other zero-shot models.

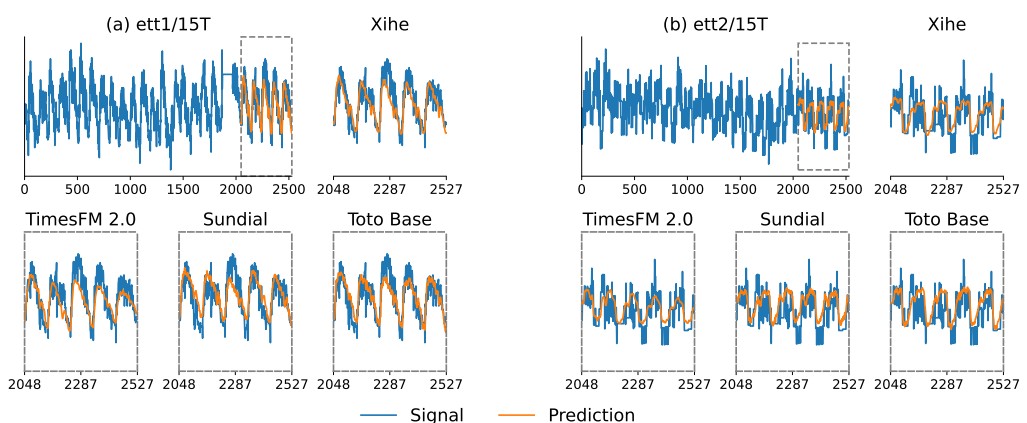

Figure 10: Examples of forecasts comparison from GIFT-Eval benchmark. For each sample, we provide both the full context and **Xihe-max** prediction, as well as the zoomed-in prediction of other zero-shot models.

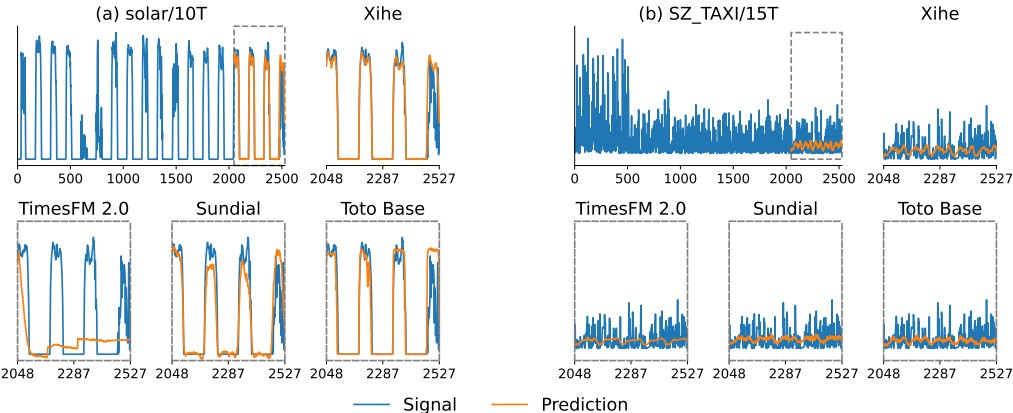

Figure 11: Examples of forecasts comparison from GIFT-Eval benchmark. For each sample, we provide both the full context and **Xihe-max** prediction, as well as the zoomed-in prediction of other zero-shot models.

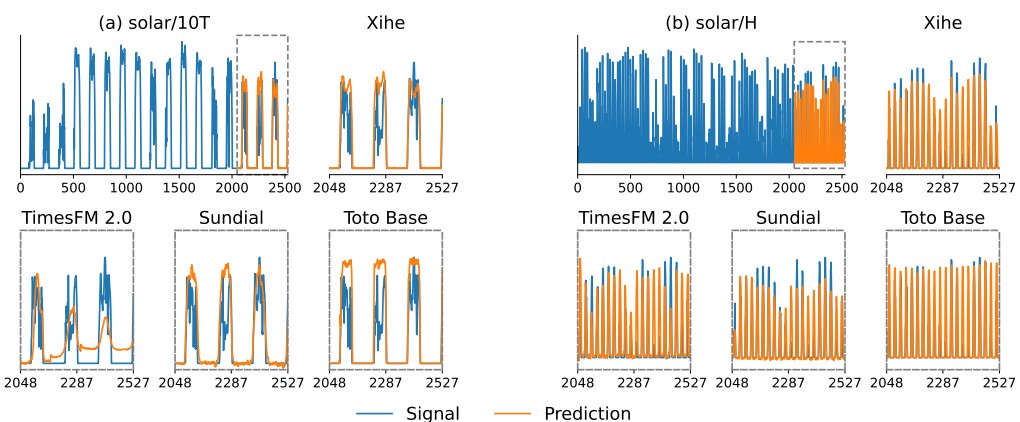

Figure 12: Examples of forecasts comparison from GIFT-Eval benchmark. For each sample, we provide both the full context and **Xihe-max** prediction, as well as the zoomed-in prediction of other zero-shot models.

# G  STATEMENT FOR LARGE LANGUAGE MODELS USAGE

Large Language Models is only used to polish the writing and does not change the author's intention.

