# OpenReview forum: "Xihe: Scalable Zero-shot Time Series Learner via Hierarchical Interleaved Block Attention"
_ICLR.cc/2026/Conference — Submitted to ICLR 2026_

### Official Review · Reviewer_vJ61 · 2025-10-29

**Soundness:** 3
**Presentation:** 3
**Contribution:** 3
**Rating:** 6
**Confidence:** 3

**Summary:**

This paper introduces Xihe, a new family of Time Series Foundation Models  designed for zero-shot forecasting. The central innovation is the Hierarchical Interleaved Block Attention  architecture. The authors argue that existing TSFMs, often adapted from NLP, fail to adequately capture the multi-scale temporal dependencies inherent in time series data. HIBA addresses this by partitioning sequences into hierarchical blocks and alternating between intra-block attention and inter-block attention. The authors present a scalable family of models  trained on a large corpus of 325B time points. Experimental results on the GIFT-Eval benchmark show that Xihe models achieve state-of-the-art zero-shot performance, with the largest model  setting a new SOTA and the smallest outperforming most contemporary TSFMs, demonstrating significant parameter efficiency.

**Strengths:**

The Xihe family demonstrates state-of-the-art zero-shot performance on the comprehensive GIFT-Eval benchmark. The fact that the smallest 9.5M parameter model  outperforms the majority of existing TSFMs is a very strong result, highlighting the architectural efficiency of HIBA.

The paper successfully trains and evaluates a family of models ranging from 9.5M to 1.5B parameters. The results show a clear scaling trend where performance on both CRPS and MASE metrics improves monotonically with model size, confirming the architecture's scalability.

**Weaknesses:**

W1: The pre-training dataset is a mix of public datasets and synthetic data. While large, the contribution of the data-quality-aware mixing strategy  versus the architectural improvements is not explicitly disentangled. It's unclear how much of the performance gain comes from this curated data mix.

W2: The implementation details of Hierarchical Block Size is ambiguous. Though authors provide information about the block size in appendix, it is not clear how to configure the block size for intra-block and inter-block attention.

W3: The design of Non-causal Intra-block Attention is not well-motivated. Since the ablation study shows that the causal intra-block attention is only slightly worse than the non-causal one, and the casual attention naturally fits the time series data and is more efficient for inference.

W4: There is clear typo in equation 5, the definition of b, m, B, M is ambiguous.

**Questions:**

listed in weakness

---

> ### Author Response · Authors · 2025-11-22
> **Response to Reviewer vJ61**
>
> We sincerely appreciate Reviewer vJ61 for considering our work is novel and solid, and your comments for improvement are professional and constructive. Below, we provide a detailed response to address your concern:
>
> >W1:The pre-training dataset is a mix of public datasets and synthetic data. While large, the contribution of the data-quality-aware mixing strategy versus the architectural improvements is not explicitly disentangled. It's unclear how much of the performance gain comes from this curated data mix.
>
> Thank you for your insightful feedback and suggestions. We conduct ablation studies with Xihe-tiny to quantify the respective contributions of data quality improvement and model architecture innovation to overall performance gains. First, we fix the model architecture and compare two data mixing strategies: (1) uniform mixing strategy and (2) data-quality-aware mixing strategy. This allows us to isolate and measure the performance gain attributable to the data mixing strategy. Then, we fix the data mixing strategy and compare different model architectures to quantify the contribution of architectural innovations.
>
> The experimental results are summarized in the table below and have been updated in the appendix D.5 of revised manuscript. They show that, under a fixed architecture, data-quality-aware mixing strategy leads to an average reduction of 0.0105/0.0045 in MASE/CRPS. In contrast, under a fixed data mixing strategy, architectural innovations yield an average reduction of 0.0125/0.0055 in MASE/CRPS. **These quantitative results indicate that the performance improvement brought by model architecture innovation is substantially larger than that achieved by data-quality-aware mixing strategy**.
>
> | MASE | uniform | data-quality-aware |
> | :--- | :--- | :--- |
> | Xihe-tiny | 0.774 | **0.766** |
> | Standard attn | 0.789 | 0.776 |
>
> | CRPS | uniform | data-quality-aware |
> | :--- | :--- | :--- |
> | Xihe-tiny | 0.542 | **0.538** |
> | Standard attn | 0.548 | 0.543 |
>
> > W2:The implementation details of Hierarchical Block Size is ambiguous. Though authors provide information about the block size in appendix, it is not clear how to configure the block size for intra-block and inter-block attention.
>
> Thank you for your concerns regarding Hierarchical Block Size. We provide the pseudocode to illustrate the parameter configuration of the HIBA block size for intra-block and inter-block attention and the execution workflow, in order to help readers better understand the working mechanism of the HIBA architecture. **This pseudocode has been updated in the of Algorithm 1 of Appendix A of the revised manuscript.**
>
> >W3:The design of Non-causal Intra-block Attention is not well-motivated. Since the ablation study shows that the causal intra-block attention is only slightly worse than the non-causal one, and the casual attention naturally fits the time series data and is more efficient for inference.
>
> For Xihe-base we retrained and evaluated it five times and updated the confidence intervals of the ablation results in Appendix D.2 of the revised manuscript. In addition, we conducted ablation studies for the Xihe-tiny, and the corresponding results have been updated in the Appendix D.4 of revised manuscript. **Across model scales, the ablation results consistently show that the non-causal intra-block attention setting outperforms the causal one.** Therefore, we adopt the non-causal  intra-block attention in our HIBA architecture.
>
> >W4: There is clear typo in equation 5, the definition of b, m, B, M is ambiguous.
>
> Thank you for pointing this out. We corrected the typo error in Equation 5 and updated its formulation description in line 229 of revised manuscript to make the variable definitions clearer. Equation 5 describes two equivalent subscript notation for the hidden state representation $h$. $b$ denotes $b$-th token within divided blocks and m denotes $m$-th divided blocks. $h_{b, m}$ denote b-th tokens within the m-th block, and $h_{(m-1)\times B + b}$ denote (m-1)\times B + b}-th token. The two notations are equivalent.

---

### Official Review · Reviewer_sK1Q · 2025-10-29

**Soundness:** 2
**Presentation:** 3
**Contribution:** 2
**Rating:** 2
**Confidence:** 5

**Summary:**

The paper proposes a time series foundation model (TSFM) architecture, that is claimed to establish new state-of-the-art in the popular GIFTEval benchmark. Specifically, it proposes Hierarchical Interleaved Block Attention (HIBA) in the Transformer architecture, which employs intra- and inter-block attention to capture multi-scale dependencies more effectively. The paper attempts to improve the state-of-the-art in ZeroShot (ZS) TS forecasting, which is an important problem in the time series domain.

**Strengths:**

The paper has the following strengths:
1. The presentation is clear, and understandable.
2. The experimental evaluation is performed on a well-established leaderboard (GIFT) which comprises of a significant amount of univariate time series evaluation data.
3. Ablations studies are reported.

**Weaknesses:**

However, the paper has the following weaknesses:
1. The concept of intra- and inter-block attention is not new. In the TSFM literature, similar concepts were proposed in the TSMixer[1] and TTM[2] papers, where they employed mixing instead of attention. How are intra- and inter-block attention conceptually different from intra- and inter-patch mixing?
2. The concept of varying the block length is not novel as well. The TTM paper proposed something which authors called as "Adaptive Patching". How is that different from the proposed variable block length?
3. Can the proposed model handle multivariate data, and model inter-channel correlations?
4. Can the authors benchmark the model on multivariate benchmarks such as the FEV leaderboard?
5. The authors claim to have established a SOTA score in GIFT, however, the leaderboard has much better SOTA numbers than the reported ones. Hence, this claim should be modified.
6. What the values of the multi-horizons? Only 96 and 768? Why not more?
7. The author(s) have claimed several times that " (TSFMs) has been propelled bymigrating architectures from language models". This statement is not entirely true. While there are some models which are adopted from language, there are many more novel models with significant modifications to handle multivariate time series data.
8. What are some of the weaknesses of the proposed model? When does it fail?
9. Were the entire 300+ Billion time points used in the pre-training? How did the authors mitigate extreme bias in the LOTSA dataset? Some of the LOTSA datasets have 15B time points, and some have 1000 time points.
10. "subsets of the training datasets from Chronos (Ansari et al., 2024a)" -- which subset?
11. "the LOTSA datasets from Moirai" -- doesn't it have GIFTEval data in it?
12. "synthetic time series generated using a procedure inspired by KernelSynth" -- what do the authors mean by inspired by KernelSynth? Is it a modified KernelSynth? The details are not provided, e.g., How many kernels? What kernels?

**Questions:**

See weakness.

---

> ### Author Response · Authors · 2025-11-22
> **Response to Reviewer sK1Q (Part 1)**
>
> We sincerely thank you for your insightful feedback and suggestions. Below, we provide a detailed response to address your concern:
>
> >Q1:The concept of intra- and inter-block attention is not new. In the TSFM literature, similar concepts were proposed in the TSMixer[1] and TTM[2] papers, where they employed mixing instead of attention. How are intra- and inter-block attention conceptually different from intra- and inter-patch mixing?
>
> We appreciate the reviewer’s engagement with foundational works. The concept of intra- and inter-block attention is conceptually different from intra- and inter-patch mixing in TSMixer and TTM. In TSMixer, the inter patch mixer module employs a shared MLP to mix time dimension and the intra patch mixer block’s shared MLP layer mixes the dimensions of the feature dimension, both operations are static mixing processes with shared MLP. In contrast, our HIBA architecture split time dimension into different blocks, and the intra and inter modules apply dynamic attention within and across blocks, enabling the model to capture local information within each block as well as global dependencies across blocks in time dimension. Our HIBA architecture fundamentally rethinks hierarchical temporal modeling through three conceptual divergences from TSMixer/TTM’s mixing-based paradigm.
>
>  * **Mechanistic Distinction: Temporal Order**
>
>     *    **TSMixer/TTM Intra-patch Mixing**: Applies MLPs that collapse temporal ordering within segments.
>     *    **HIBA Intra-block Attention**: Uses token-level adaptive self-attention to explicitly preserve sequential dynamics. This allows modeling ordered temporal dependencies (e.g., lagged correlations, phase relationships) that mixing discards—critical for non-stationary series where temporal contiguity informs evolution.
>
>  * **Hierarchical Dynamics: Static Projections vs. Adaptive Cross-Scale Modeling**
>
>     *    **Inter-patch Mixing (TTM/TSMixer)**: Relies on fixed linear projections for cross-segment interactions, forcing global patterns into predetermined feature subspaces.
>
>     *    **HIBA Inter-block Attention**: Employs attention over dynamically block token representations, enabling adaptive refinement of global context based on localized intra-block outputs. This hierarchy (intra-block → inter-block) captures co-evolving multi-scale dependencies directly resolving the "constrained multi-scale capture" limitation noted in our introduction.
>
>  * **Generalization Mechanism: Contextualized Transfer vs. Homogeneous Abstraction**
>
>     *    **Mixing’s Limitation**: Aggregates features via shared MLP operations, creating over-smoothed representations ill-suited for datasets with divergent sampling strategies.
>
>     *    **HIBA’s Advantage**: Attention’s contextual token interactions maintain dataset-specific temporal signatures during zero-shot transfer. This enables Xihe’s state-of-the-art performance (e.g., 6.0%/15.9% zero-shot MASE/CRPS performance gain over TTM-R2-Finetuned on GIFT-Eval benchmark), as sparse attention selectively amplifies salient cross-dataset patterns without reverting to spatial-like biases.
>
> We have also added citations to TTM and TSMixer, along with a comparative discussion of their model concepts in line 219 of the revised manuscript.
>
> [1] Ekambaram, Vijayabharathi et al. “TSMixer: Lightweight MLP-Mixer Model for Multivariate Time Series Forecasting.” Proceedings of the 29th ACM SIGKDD Conference on Knowledge Discovery and Data Mining (2023)
>
> [2]Ekambaram, Vijay et al. “Tiny Time Mixers (TTMs): Fast Pre-trained Models for Enhanced Zero/Few-Shot Forecasting of Multivariate Time Series.” ArXiv abs/2401.03955 (2024)
>
> >Q2:The concept of varying the block length is not novel as well. The TTM paper proposed something which authors called as "Adaptive Patching". How is that different from the proposed variable block length?
>
> We appreciate the reviewer’s engagement with foundational works. The adaptive patching proposed in TTM paper is different from hierarchical block size of HIBA. In TTM backbone, adaptive patching is achieved by partitioning, exchanging, and merging segment across the temporal and feature dimension at diferent segment size. Such mechanism collapse temporal ordering at different scale with fixed linear projections and **exchanging between temporal and feature dimension**. HIBA partition the time dimension at different scale in each layer and use token-level self-attention to explicitly preserve sequential dynamics within and across blocks. The feature and **temporal dimension remain independent throughout the temporal information exchange process**. This hierarchy setting **captures co-evolving multi-scale dependencies by cross-scale modeling across tokens with temporal order.**

---

> > ### Author Response · Authors · 2025-11-22
> > **Response to Reviewer sK1Q (Part 2)**
> >
> > > Q3:Can the proposed model handle multivariate data, and model inter-channel correlations?
> >
> > We thank the reviewer for bringing up the question of multivariate capability and inter-channel correlation modeling. The current version of our foundation model was developed and evaluated on univariate time series. While the architecture could be extended to accept multivariate inputs by treating channels as distinct feature dimensions, we did not experiment with such data in this work.
> >
> > We acknowledge that modeling inter-channel dependencies is an important aspect for real-world scenarios. We have added a note in the conclusion section( line 449) in revised manuscript outlining how our attention-based framework could be adapted to multivariate inputs (e.g., shared embeddings per time step, channel-specific encodings) and will investigate this direction as part of future work.
> >
> > >Q4:Can the authors benchmark the model on multivariate benchmarks such as the FEV leaderboard?
> >
> > We thank the reviewer for suggestion on FEV leaderboard.  We have added results on the FEV benchmark, as shown in the table below. The evaluation metric is the win rate, where higher values indicate better performance. Our model remains highly competitive in overall performance.
> >
> > | model_name | WQL win rate | MASE win rate |
> > | :--- | :--- | :--- |
> > | xihe_max | **71.125** | **65.8125** |
> > | Toto-1.0 | 68.125 | 64.5 |
> > | TabPFN-TS | 65.09375 | 56.843 |
> > | Moirai-2.0 | 63.5625 | 59.3125 |
> > | Chronos-Bolt | 61.5625 | 58.4375 |
> > | Sundial-Base | 42.46875 | 50.09375 |
> > | Stat. Ensemble | 40.34374 | 46.03125 |
> > | AutoARIMA | 36.09375 | 34.46875 |
> > | AutoETS | 29.03125 | 31.90625 |
> > | AutoTheta | 23.9375 | 32 |
> > | Seasonal Naive | 17.15625 | 19.28125 |
> > | Naive | 12.03125 | 17.59375 |
> > | Drift | 8.4375 | 14.62500 |
> >
> > >Q5: The authors claim to have established a SOTA score in GIFT, however, the leaderboard has much better SOTA numbers than the reported ones. Hence, this claim should be modified.
> >
> > We thank the reviewer for pointing out the discrepancy between our claim and the current GIFT leaderboard results. Our SOTA claim was based on the leaderboard status at the time of our paper submission (2025-9-24). We agree that in the current leaderboard, higher scores have since been reported by other models. Notably, Xihe-max remains among top three zero-shot models on the current GIFT-Eval leaderboard. In the revised manuscript, we have updated the statement in the abstract to clarify this point.
> >
> > >Q6: What the values of the multi-horizons? Only 96 and 768? Why not more?
> >
> > We thank the reviewer for the concern on the sensitivity analysis on multihead prediction. The corresponding experimental results are updated in the Table 1 left and Appendix D.3. We conducted ablation studies for Xihe-base with more than two prediction heads. The results show that adding too many prediction heads does not yield further performance gains, suggesting that combination of long prediction head and short prediction head is sufficient to maintain strong predictive performance.
> >
> > >Q7:The author(s) have claimed several times that " (TSFMs) has been propelled bymigrating architectures from language models". This statement is not entirely true. While there are some models which are adopted from language, there are many more novel models with significant modifications to handle multivariate time series data.
> >
> > We thank the reviewer for their valuable comment and for pointing out the need for clarification regarding our statement on the origins of TSFMs. Our original wording may have given the impression that TSFMs are predominantly migrated directly from language model architectures. We acknowledge that, although a substantial proportion of TSFMs—particularly those developed for univariate time series—draw upon Transformer‑based designs originally devised for natural language processing, a considerable body of work has introduced novel architectures and substantial modifications specifically designed to address the unique challenges of multivariate time series analysis.
> >
> > In response, we have revised the relevant sentences in introduction (line 41) of revised manuscripts to clarify that TSFMs have benefited from both the migration of successful design principles from language models and the development of architecture innovations unique to time series data. We have also added citations to representative works that these architecture innovations unique to time series data (e.g.TTM,Tirex), as well as those adapted from NLP (e.g. Chronos, TimesFM). This revision ensures that the manuscript more accurately reflects the breadth of architectural influences in TSFM development.

---

> > ### Comment · Reviewer_sK1Q · 2025-11-27
> > **Question regarding intra-block self-attention**
> >
> > The authors said that _"HIBA Intra-block Attention: Uses token-level adaptive self-attention to explicitly preserve sequential dynamics. This allows modeling ordered temporal dependencies (e.g., lagged correlations, phase relationships) that mixing discards—critical for non-stationary series where temporal contiguity informs evolution."_.
> > The authors also said in the paper that _"In intra-block attention, a non-causal multi-head self-attention (MSA^non-causal) is applied to the hidden representations within each block."_.
> > Can the authors please explain how a non-causal self-attention preserve temporal order?

---

> > ### Comment · Reviewer_sK1Q · 2025-11-27
> > **Question regarding shared weights**
> >
> > The authors said that _"In TSMixer, the inter patch mixer module employs a **shared MLP** to mix time dimension and the intra patch mixer block’s **shared MLP** layer mixes the dimensions of the feature dimension, both operations are **static mixing processes with shared MLP**. In contrast, our HIBA architecture split time dimension into different blocks, and the intra and inter modules apply **dynamic attention** within and across blocks, enabling the model to capture local information within each block as well as global dependencies across blocks in time dimension."_.
> >
> > - Can the authors please explain what they mean by "static mixing"?
> > - Can the authors please explain what they mean by "dynamic attention"?
> > - Can the authors please explain how one self-attention layer does _not_ use shared Q, K, V matrices?

---

> > > ### Comment · Reviewer_sK1Q · 2025-11-27
> > > **Is the accuracy coming from model improvement or data curation?**
> > >
> > > The authors mentioned that _"Then, a **human expert** would consider these indicators together with visual inspection of the sampled sequences and assess the forecastability (classified as high or low) and noise level (high, medium, or low) of each dataset. We rely on human assessment because, given the great diversity of time-series patterns across rich real-world data, no single statistical metric can adequately capture overall forecastability."_.
> > >
> > > - How is this method of data curation scalable?
> > > - It seems like the data curation involved meticulously chosen methods, hyperparameters, and human in the loop involvement. Given this, if the authors train any other existing SOTA model(s) on the same _curated_ data, the other models might give similar results. What can the research community infer from this? Is the competitive accuracy on the benchmarks coming from rigorous data curation or, the claimed model improvements?

---

> ### Author Response · Authors · 2025-11-22
> **Response to Reviewer sK1Q (Part 3)**
>
> >Q8:What are some of the weaknesses of the proposed model? When does it fail?
>
> We thank the reviewer for the suggestion for the weaknesses of Xihe. We list following three weakness and hope this can address your concerns.
>  * Our current model targets univariate time-series forecasting. While it can be extended to multivariate settings, it does not explicitly model inter-series dependencies.
>  * The present work focuses on pretraining for forecasting task and does not yet support other downstream tasks such as classification and anomaly detection.
>  * The current implementation does not support the integration of external information (exogenous covariates).
>
> We have add the weakness disscusion in conslusion section of revised manuscript and leave it as future exploration.

---

> > ### Author Response · Authors · 2025-11-22
> > **Response to Reviewer sK1Q (Part 4)**
> >
> > >Q9:Were the entire 300+ Billion time points used in the pre-training? How did the authors mitigate extreme bias in the LOTSA dataset? Some of the LOTSA datasets have 15B time points, and some have 1000 time points.
> >
> > >Q10:"subsets of the training datasets from Chronos (Ansari et al., 2024a)" -- which subset?
> >
> > >Q11:"the LOTSA datasets from Moirai" -- doesn't it have GIFTEval data in it?
> >
> > >Q12:"synthetic time series generated using a procedure inspired by KernelSynth" -- what do the authors mean by inspired by KernelSynth? Is it a modified KernelSynth? The details are not provided, e.g., How many kernels? What kernels?
> >
> > We apologize for not clearly describing our pretraining datasets in detail. Below, we describe our data sources, synthetic data generation algorithm, and data mixing strategy to address your questions. In the revised version of the paper, additional details on the pre-training dataset are provided in Appendix E.
> >
> > Real-world Data Sources:
> > Our real-world datasets consist of collections from Chronos and the LOTSA dataset, comprising over 300 billion data points.
> > To accurately evaluate our model’s zero-shot performance on the GIFT-Eval benchmark, we removed all datasets appearing in GIFT-Eval from the final training corpus to avoid data leakage.
> >
> > Synthetic Data:
> > Generally, We expand the KernelSynth algorithm proposed in Chronos [1] in the following aspects:
> > 1. Extend length: We increase the maximum generated sequence length from 1024 to 4096
> > 2. Extend kernel: We add ExpSineSquared kernels with small length_scale (0.1) to generate spike signals. We also adjust the periodic component of the ExpSineSquared kernel, which we describe in detail in the next point.
> > 3. Kernel Sample Strategy: Compared with the original strategy of randomly sampling multiple kernels, we impose a constraint that the sampled kernels must include either a periodic component (ExpSineSquared) or a smooth trend component (DotProduct or RBF), ensuring that the generated sequences remain forecastable. For the periodic kernels, we further restrict the number of periodic components to at most two. The period $p_1$ of the first component is randomly selected from $\{4, 7, 12, 24, 52, 60, 96, 144, 168, 288, 360, \mathrm{rand}(4, 2016)\}$, where $\mathrm{rand}(n, m)$ denotes uniform sampling of an integer from $[n, m]$. If a second periodic component is included, its period is set to $k \cdot p_1$, where $k$ is randomly sampled from $\{7, 52, \mathrm{rand}(4, 100)\}$
> > These extensions make the data generated by KernelSynth more closely resemble real-world time-series characteristics, even under greater diversity.
> >
> > Data Mixture Strategy:
> > We adopt a data-quality-based data mixture strategy. For real-world datasets, we first assess the data quality of each dataset. Specifically, we first sample multiple sequences from each dataset and compute indicators such as ACF, trend/seasonal strength after STL decomposition, Hurst exponent, and spectral entropy, which characterize periodicity, trend, and noise levels. Then, a human expert would consider these indicators together with visual inspection of the sampled sequences and assess the forecastability (classified as high or low) and noise level (high, medium, or low) of each dataset. We rely on human assessment because, given the great diversity of time-series patterns across rich real-world data, no single statistical metric can adequately capture overall forecastability.
> >
> > We subsequently divide all real-world datasets into five groups according to the assessed labels (forecastability-noise level). The groups are: high–low, high–medium, high–high, low–low, and a combined low–medium/high category, where the medium and high noise levels are merged due to their similar characteristics under low forecastability. These five groups are assigned sampling probabilities of 40%, 20%, 10%, 7%, and 3%, respectively, during training. The synthetic data is assigned sampling probability of 20%, as our extended KernelSynth algorithm guarantees a certain level of forecastability. Within each group, all datasets are sampled with equal probability to ensure that differences in dataset size do not introduce sampling imbalance. This allocation ensures that the model is trained predominantly on high-quality data while still maintaining strong generalization ability.
> >
> > [1] Ansari, Abdul Fatir, et al. "Chronos: Learning the language of time series." arXiv preprint arXiv:2403.07815 (2024).

---

> ### Author Response · Authors · 2025-12-02
> **Response to Reviewer sK1Q (Question regarding intra-block self-attention)**
>
> >The authors said that "HIBA Intra-block Attention: Uses token-level adaptive self-attention to explicitly preserve sequential dynamics. This allows modeling ordered temporal dependencies (e.g., lagged correlations, phase relationships) that mixing discards—critical for non-stationary series where temporal contiguity informs evolution.".
> The authors also said in the paper that "In intra-block attention, a non-causal multi-head self-attention (MSA^non-causal) is applied to the hidden representations within each block.".
> Can the authors please explain how a non-causal self-attention preserve temporal order?
>
> ---
>
> Thank you for the interest on  how non-causal self-attention preserve temporal order. Here's how a non-causal intra-block preserves temporal order:
>
> 1.  **Global Sequential Dynamics / Temporal Order:** This refers to the chronological relationship between different token of time steps or "blocks" across the entire time series (e.g., the token at time $t_1$ precedes the token at $t_2$). Our claim that HIBA "explicitly preserves sequential dynamics" is primarily a contrast to methods like TSMixer or TTM, which might obscure or lose this global order through patch-mixing. HIBA, by design, **maintains the original sequential arrangement of its patched tokens via positional embedding** and each token is inherently encoded with sequential information which are standard in Transformer architectures.
>
> 2.  **Causality within Attention Computation:** This refers to whether a specific token, during its attention calculation, is permitted to "see" information from future tokens within its local context (i.e., its block).
>
> With this distinction in mind, we can now elaborate:
>
> #### How HIBA Preserves Global Temporal Order
>
> *   **Ordered Patch Tokenization:** The time series is first segmented into patches and tokenized. This sequence of tokens, $h_{1:n}$, strictly follows the original temporal order.
> *   **Positional Information:** HIBA architectures rely on **Positional Encodings** to embed the absolute or relative position of each token into its representation. This ensures the model is always aware of the sequence order during attention computations.
>
> #### How Non-Causal Intra-Block Attention Operates without Destroying Temporal Order
>
> *   **"Non-Causal" Refers to Information Flow, Not Order Destruction:** Within intra-block attention, the term "non-causal" means that any given token can attend to **all** other tokens within its block—including past, present, and future tokens *within that local window*. This enables a rich, bidirectional fusion of local information. Such a mechanism is highly effective for learning complex local interdependencies, such as capturing local periodicities or trend-reversal patterns contained within a block.
> *   **Order is Preserved via Positional Information:** Even though the attention is non-causal, the model remains aware of the relative order of tokens via their **positional encodings**. Therefore, the model can learn that the relationship between token $k$ and token $k+1$ is different from the relationship between token $k$ and token $k-1$. Non-causality simply removes the hard constraint of unidirectional information flow, allowing for more comprehensive local pattern learning.
>
> **In summary, the concepts of "preserving temporal order" and "non-causal self-attention" are not contradictory in our framework.** The former refers to the model's ability to recognize and utilize the global sequential structure, which is maintained throughout. The latter describes a local, bidirectional information fusion process designed for enhanced feature extraction within a block. Positional encodings act as the bridge, allowing the model to leverage non-causal local attention while remaining fully aware of the temporal order.

---

> > ### Author Response · Authors · 2025-12-02
> > **Response to Reviewer sK1Q (Question regarding shared weights) (Part 1)**
> >
> > > The authors said that "In TSMixer, the inter patch mixer module employs a shared MLP to mix time dimension and the intra patch mixer block’s shared MLP layer mixes the dimensions of the feature dimension, both operations are static mixing processes with shared MLP. In contrast, our HIBA architecture split time dimension into different blocks, and the intra and inter modules apply dynamic attention within and across blocks, enabling the model to capture local information within each block as well as global dependencies across blocks in time dimension.".
> > > * Can the authors please explain what they mean by "static mixing"?
> > ----
> >
> > The term "static mixing," as used in the context of TSMixer, refers to its core mechanism of feature interaction, which relies primarily on **shared Multi-Layer Perceptrons (MLPs)**. Specifically:
> >
> > *   **Temporal Mixing (Inter-patch):** TSMixer employs a shared MLP to mix information across different time steps (or patches). The weights of this MLP are **fixed and invariant** across all temporal positions in the sequence once training is complete.
> > *   **Feature Mixing (Intra-patch):** A separate shared MLP is used to mix information across different feature dimensions within each time step. The weights of this MLP are also **fixed and invariant** across all feature dimensions.
> >
> > The "static" nature of this approach is characterized by:
> >
> > 1.  **Fixed Weights:** Once trained, the weight matrices of the MLPs are constant.
> > 2.  **Input-Agnostic Transformation:** The MLPs apply the exact same linear transformations and non-linear activations to any given input. The mixing operation does not adapt based on the **content** of the input sequence. The pattern of interaction is predetermined.
> > 3.  **Lack of Contextual Awareness:** The manner in which information from one time step (or feature) is combined with another is dictated by the globally learned MLP weights, not by the **dynamic relationships** between specific elements within the current input sequence.
> >
> > Therefore, "static mixing" implies a pre-defined, globally consistent method of feature combination that does not dynamically change in response to input content.
> >
> > ---
> > > * Can the authors please explain what they mean by "dynamic attention"?
> > ---
> > In contrast to "static mixing," "dynamic attention" refers to the ability of the self-attention mechanism to **adaptively adjust the information mixing process based on the actual content and context of the input data.**
> >
> > The "dynamic" aspect is manifested in several key properties:
> >
> > 1.  **Content-Dependent Weights:** The attention weights are not fixed parameters. They are computed on-the-fly for each input by measuring the similarity (e.g., via dot product) between a **Query (Q)** and a set of **Keys (K)**. These Q, K, and **Value (V)** vectors are themselves derived directly from the input content.
> > 2.  **Adaptive Aggregation:** Each Query dynamically determines which parts of the input sequence (represented by the Keys) to "attend" to and to what degree. For instance, a Query representing a sudden peak in a sequence might assign higher attention scores to Keys corresponding to related precursor events within that same sequence.
> > 3.  **Context-Aware Interaction:** Attention scores are calculated for every Query-Key pair, enabling the model to flexibly and selectively draw the most relevant information from other tokens when constructing the output representation for a specific token. This relevance is re-evaluated for every input.
> >
> > Consequently, "dynamic attention" facilitates a highly flexible and adaptive information fusion mechanism, tailored to the specific context of each input, which stands in stark contrast to the fixed transformation patterns of an MLP.

---

> > > ### Author Response · Authors · 2025-12-02
> > > **Response to Reviewer sK1Q (Question regarding shared weights) (Part 2)**
> > >
> > > > * Can the authors please explain how one self-attention layer does not use shared Q, K, V matrices?
> > > ----
> > > A common point of confusion is whether a self-attention layer uses shared Q, K, and V matrices. It is critical to distinguish between shared *projection matrices* and the resulting *dynamic interactions*.
> > >
> > > *   **Shared Projection Matrices:** **Yes, within a single self-attention layer (or head), the projection matrices ($W_Q, W_K, W_V$) used to transform the input tokens into Query, Key, and Value vectors are indeed shared.** For an input sequence of $N$ tokens, $\{x_1, x_2, ..., x_N\}$, the same learned weight matrices $W_Q, W_K, W_V$ are applied to every token $x_i$ to generate its corresponding vectors $q_i, k_i, v_i$. In this sense, the parameters $W_Q, W_K, W_V$ are shared across all positions in the sequence.
> > >
> > > *   **The Dynamic Interaction Process:** The "dynamic" nature of self-attention arises not from the projection matrices themselves, but from the subsequent computational steps:
> > >     1.  **Unique Q, K, V Vectors:** Although the projection matrices are shared, the resulting vectors ($q_i, k_i, v_i$) are unique for each token $x_i$, as they are a direct function of its specific content.
> > >     2.  **Unique Attention Scores:** For each query $q_i$, a unique set of attention scores, $\alpha_{ij}$, is computed by comparing it against all keys $k_j$ in the sequence. This scoring is entirely content-driven and context-dependent.
> > >     3.  **Dynamically Weighted Output:** The final output representation for each token is a weighted sum of all Value vectors, where the weights ($\alpha_{ij}$) have been dynamically computed. This ensures the information aggregation is tailored to the specific token's role and context.
> > >
> > > In summary, when contrasting TSMixer's "static mixing" with "dynamic attention," the key distinction is not whether projection matrices are shared. The critical difference is that **the self-attention mechanism performs a content-dependent, adaptive computation to determine how information is mixed, whereas an MLP applies a fixed, input-agnostic transformation.**

---

> ### Author Response · Authors · 2025-12-02
> **Response to Reviewer sK1Q (Is the accuracy coming from model improvement or data curation?)**
>
> >The authors mentioned that "Then, a human expert would consider these indicators together with visual inspection of the sampled sequences and assess the forecastability (classified as high or low) and noise level (high, medium, or low) of each dataset. We rely on human assessment because, given the great diversity of time-series patterns across rich real-world data, no single statistical metric can adequately capture overall forecastability.".
> > * How is this method of data curation scalable?
> ----
>
> Thank you for the reviewer on the interest on the data curation of our model. The training dataset contains both harmonic and non-harmonic time series. Our human expert assessment procedure is applied only to the non-harmonic series, because for these sequences the strengths of trend and seasonal components cannot be reliably estimated. Moreover, such non-harmonic series account for only a small fraction of the overall dataset. In contrast, for harmonic series, Indicators can be effectively computed and the sequences can then be automatically classified based on thresholding. The procedure for determining sequence forecastability can be further scaled up by training a reward model to automate this process. We leave this direction as an avenue for future work.
>
> ----
> >* It seems like the data curation involved meticulously chosen methods, hyperparameters, and human in the loop involvement. Given this, if the authors train any other existing SOTA model(s) on the same curated data, the other models might give similar results. What can the research community infer from this? Is the competitive accuracy on the benchmarks coming from rigorous data curation or, the claimed model improvements?
> ----
>
> We demonstrate the performace gain of our method from two perspectives: (i) **The proposed data curation strategy can also provide performance gains for other state-of-the-art models, such Chronos-bolt and** (ii)  **the performance improvement brought by model architecture innovation is substantially larger than that achieved by data-quality-aware mixing strategy.**
>  * We conduct ablation studies with Xihe-tiny to disentangle the respective contributions of data quality improvement and architectural innovation to the overall performance gains. Specifically, we first fix the model architecture and compare two data mixing strategies: (i) a uniform mixing strategy and (ii) a data-quality-aware mixing strategy. This design allows us to isolate and quantify the performance improvement attributable solely to the data-quality-aware mixing. Next, we fix the data mixing strategy and vary the model architecture, thereby measuring the effect of architectural innovations. The results, summarized in the table below and updated in Appendix D.5 of the revised manuscript, show that, under a fixed architecture, the data-quality-aware mixing strategy yields **an average reduction of 0.0105/0.0045 in MASE/CRPS**. In contrast, under a fixed data mixing strategy, architectural innovations lead to **a larger average reduction of 0.0125/0.0055 in MASE/CRPS**. **These findings quantitatively demonstrate that the performance gains arising from architectural innovations are substantially greater than those obtained from the data-quality-aware mixing strategy.**
>  * Building upon the architecture of Amazon’s state-of-the-art foundation model Chronos-bolt-base(205M), we retrained the model Chronos-bolt-base-retrained(205M) from the scratch using our current datasets and data curation strategy. As shown in the following results, the retrained model, Chronos-bolt-base-retrained(205M), **outperforms the original Chronos-Bolt on the GIFT-Eval leaderboard**, demonstrating that the effectiveness of our data curation approach and **data curation strategy can apply on other state-of-the-art model. However, its performance still falls short of our Xihe-lite(94M), highlighting the superiority of our model architecture.**
>
> | | CRPS | MASE |
> |---|---|---|
> | Chronos-bolt-base(205M) | 0.574 | 0.808 |
> | Chronos-bolt-base-retrained | 0.550 | 0.760 |
> | Xihe-lite(94M) | **0.508** | **0.729** |

---

### Official Review · Reviewer_zF4J · 2025-11-03

**Soundness:** 2
**Presentation:** 4
**Contribution:** 2
**Rating:** 2
**Confidence:** 4

**Summary:**

This paper introduces Xihe, a family of Time Series Foundation Models (TSFMs), for the task of zero-shot forecasting. Their motivation is that existing TSFMs are often adapted directly from NLP/CV and therefore struggle to effectively capture the inherent multi-scale temporal dependencies, which are important for time series. To address this, the paper proposes the Hierarchical Interleaved Block Attention (HIBA) mechanism. HIBA hierarchically partitions the input sequence into blocks of varying granularity across different layers. It alternates between two types of attention: 1) intra-block attention, which models local dependencies within each block, and 2) inter-block attention, which models global dependencies across all patches. The resulting models ranges from 9.5M to 1.5B parameters, which are pretrained on a mixture of public data and synthetic data from existing TSFMs. The results on the GIFT-Eval benchmark show an outstanding performance. The paper also demonstrates that the proposed family of models follows clear scaling laws.

**Strengths:**

1. The paper presents the results on the GIFT-Eval benchmark, instead of only using the seven small LTF datasets.
2. The writing is very good and easy to follow. However, the authors tend to use ";" too often. For example, “strong zero-shot capabilities; Although Moirai”, you should replace “;” with “.”. Some other grammar errors:
- Some baseline methods are misspelled: “Dlinear” → “DLinear”, “PatchTsT” → “PatchTST”
- Line 84: "combining public available datasets" -> "combining publicly available datasets"
- Line 96: "while remaining efficiency suitable" -> "while remaining efficient and suitable"
- LIne 182: "The detailed design... are presented" -> "The detailed design... is presented".

**Weaknesses:**

1. Not enough ablations on the hierarchical structure.
2. Not enough ablations on the K prediction heads, and this part does not seem very novel/scalable.
3. Lack of pretraining details, such as dataset mixing strategies.

I feel like the paper in the current form is not ready for acceptance. My main concern is the current ablation studies are not comprehensive enough to highlight the main contribution of this paper, which is to use a hierarchical structure of interleaved intra-block and inter-block attention. However, I am willing to raise my score if you can address my concerns.

**Questions:**

1. How do the different block sizes and number of layers work? For example, when B=(3,7,21) and there are 24 layers, does that mean in each layer, I sequentially process the input with 3, then 7, and finally 21?
2. Which prediction heads do you use? You only mentioned a total of K prediction heads. From what I understand, if the task is to predict 96 steps, then you only use the predictions from the head that predicts 96 steps, and ignore the predictions from all the other heads. In this case, how well does the model scale to a different forecast horizon not supported by any of the heads? Can you show some sensitivity analysis on whether having too many prediction heads will negatively affect the model performance?
3. Can you add some sensitivity analysis on the block sizes? It is always (3, 7, 21). Have you tried more than three block sizes and also different numbers? Do these numbers need to be in an increasing order? In the ablation study, does “w/o Hierarchy (B=3)” mean that you only use a single block of block size 3? Have you tried the same block size (e.g., (3, 3, 3) or (7, 7, 7))? For the “standard attn” ablation, do you use the same number of layers or twice the number of layers (since each layer has both intra-block attention and inter-block attention)?
4. In Table 1 left, the magnitude of differences seems to be quite small. Can you run the same model with different seeds and report the confidence intervals? Is the observation consistent across different model scales?
5. Why do tiny, lite and flash all have 24 layers? The number of layers seems to be noticeably higher than other transformer-based foundation models.
6. Line 280, “we adopt a data-quality–aware mixing strategy instead of the uniform mixing commonly used in prior TSFMs”. How exactly does this work?
7. In Line 264, why is there information leakage? If you apply left padding for the context, the context will be divisible by the patch size and block size. Why would there still be information leakage from the forecast horizon? My understanding is that the acausal signals learned during the non-causal intra-block attention will impact the subsequent causal inter-block attention, so there will be some information leakage during training, but not testing.
8. Have you tried using a student-t distribution (https://arxiv.org/pdf/2410.12360, https://arxiv.org/pdf/2402.02592) instead of quantile loss as the loss function?

---

> ### Author Response · Authors · 2025-11-22
> **Response to Reviewer zF4J (Part 1)**
>
> We sincerely thank the reviewer zF4J for the positive recognition of our paper presentation and your comments for improvement are professional and constructive. Below, we provide a detailed response to address your concern:
>
> >W: 1. Not enough ablations on the hierarchical structure.
> 2. Not enough ablations on the K prediction heads, and this part does not seem very novel/scalable.
> 3. Lack of pretraining details, such as dataset mixing strategies.
> I feel like the paper in the current form is not ready for acceptance. My main concern is the current ablation studies are not comprehensive enough to highlight the main contribution of this paper, which is to use a hierarchical structure of interleaved intra-block and inter-block attention. However, I am willing to raise my score if you can address my concerns.
>
> We thank the reviewer for the concerns for ablation study and suggestions for improvement. To highlight the main contribution of our work, we have added following ablation studies addressing three concerns about hierarchical structure, K prediction heads and mixing strategies raised by the reviewer.
> * **Hierarchical structure**. 1）To clarify how different block sizes and number of layers of the HIBA architecture cooperate, we have updated the model pseudocode **Algorithm 1** in **Appendix A** of revised manuscript to help readers better understand its execution procedure. 2）To clarify the validation of the block size setting, we added the ablation study in **Table 1 left** and **Appendix D** of revised manuscript which shows that hierarchical block sizes performs better than unform block sizes, the order of block sizes has a minor impact on the final performance and the observed ablation results is consistent across model sizes. Above ablation study shows that hierarchical design of HIBA helps to better model multi-scale information in time series.
> * **K prediction heads**. 1）The detailed description of the multi prediction heads mechanism are updated in **Appendix D** of revised manuscript 2）Corresponding experimental results are updated in the **Table 1** right and **Appendix D** of revised manuscripts. The results shows that adding too many prediction heads does not yield further performance gains, suggesting that combination of long prediction head and short prediction head is sufficient to maintain strong predictive performance. The observed ablation results is consistent across model sizes.
> * **Mixing strategies**. 1) The composition of the pre‑training datasets and mixing strategies is updated in **Appendix E** of revised manuscript. 2)To explicitly disentangle the contribution of the data-quality-aware mixing strategy versus the architectural improvements, we conduct ablation studies to quantify the respective contributions of data quality improvement and model architecture innovation to overall performance gains which is updated in **Appendix D.5** of revised manuscript.

---

> > ### Author Response · Authors · 2025-11-22
> > **Response to Reviewer zF4J (Part 2)**
> >
> > > Q1:How do the different block sizes and number of layers work? For example, when B=(3,7,21) and there are 24 layers, does that mean in each layer, I sequentially process the input with 3, then 7, and finally 21?
> >
> > We thank the reviewer for the comment regarding how different block sizes and number of layers cooperate. As illustrated in Fig. 1, each HIBA block contains two sublayers—$HIBA_{intra}$ layer and $HIBA_{inter}$ layer—and operates with a single B size (e.g., B=3). **Consequently, a 24-layer stack corresponds to 12 HIBA blocks. Under the schedule B=(3,7,21), the triplet of B values is cycled four times across the stack sequentially.** To further clarify the different block sizes and number of layers of the HIBA architecture, we have updated the model pseudocode in the **Algorithm 1 of Appendix A** to help readers better understand its execution procedure.
> >
> > >Q2:Which prediction heads do you use? You only mentioned a total of K prediction heads. From what I understand, if the task is to predict 96 steps, then you only use the predictions from the head that predicts 96 steps, and ignore the predictions from all the other heads. In this case, how well does the model scale to a different forecast horizon not supported by any of the heads? Can you show some sensitivity analysis on whether having too many prediction heads will negatively affect the model performance?
> >
> > We thank the reviewer for the concern on the mechanism and sensitivity analysis on multihead prediction. The detailed description of the multi prediction heads mechanism are updated in Appendix D.3 and corresponding experimental results are updated in the Table 1 right.
> > * Suppose the prediction heads have output patch {96, 480, 768}. If the forecasting horizon of task is less than or equal to 96 steps, the model uses the prediction head with output patch 96; if the forecasting horizon of task is greater than 96 steps but less than or equal to 480 steps ,it uses the head with output patch 480;if the forecasting horizon of task exceeds 480, it uses the head with output patch 768.
> > * We further conducted ablation studies for Xihe-base with more than two prediction heads (see the table below). The results show that adding too many prediction heads does not yield further performance gains, suggesting that combination of long prediction head and short prediction head is sufficient to maintain strong predictive performance.
> >
> > | | MASE | CRPS |
> > | :--- | :--- | :--- |
> > | w/ output patch{96,480,768} | **0.717** | *0.498* |
> > | w/ output patch{96,480,600,768} | *0.718* | **0.497** |
> > | w/ output patch{96} | 0.748 | 0.537 |
> > | w/ output patch{768} | 0.720 | 0.502 |
> > | w/ output patch{96,768}(Xihe-base) | *0.718* | **0.497** |

---

> > > ### Author Response · Authors · 2025-11-22
> > > **Response to Reviewer zF4J (Part 3)**
> > >
> > > >Q3:Can you add some sensitivity analysis on the block sizes? It is always (3, 7, 21). Have you tried more than three block sizes and also different numbers? Do these numbers need to be in an increasing order? In the ablation study, does “w/o Hierarchy (B=3)” mean that you only use a single block of block size 3? Have you tried the same block size (e.g., (3, 3, 3) or (7, 7, 7))? For the “standard attn” ablation, do you use the same number of layers or twice the number of layers (since each layer has both intra-block attention and inter-block attention)?
> > >
> > > We thank the reviewer for suggestion on the sensitivity analysis on block sizes. We have conducted a supplementary ablation study for sensitivity analysis on the block sizes, the results are presented in the table below and have also been updated into the Appendix D.2 and Table 1 right in revised manuscript.
> > > * As illustrated in Fig. 1, each HIBA block contains two sublayers—HIBA_{intra} layer and HIBA_{inter} layer—and operates with a single B size (e.g., B=3). The ablation study of HIBA architecture is conducted from the following four perspectives.
> > >
> > >   * **Number of block sizes**: Relative to the hierarchical block sizes (3, 7, 21), Using too many block sizes, such as (3, 7, 21, 42), can degrade performance. Under a fixed total depth, introducing an excessive number of block sizes reduces the effective number of feature-extraction cycles within stacked layers, which in turn may diminish the model’s representational capacity.
> > >
> > >    *  **Different block size**:  Replacing hierarchical block sizes (3, 7, 21) with  (4, 8, 16) results in worse performance, suggesting that hierarchical block sizes (3, 7, 21) more effectively captures informative time-series patterns in our pretraining datasets.
> > >
> > >   *  **Inceasing order**: Compared with hierarchical block sizes (3,7,21) in HIBA,  reversing order hierarchical block sizes (21,7,3) archive comparable performance which shows that the model’s performance is relatively insensitive to the ordering of block sizes.
> > >
> > >   *  **Uniform block sizes**: Compared with hierarchical block sizes (3,7,21) in HIBA, uniform block size 3 (3,3,3) or uniform block size 7 (7,7,7) lead to performance drop which shows hierarchical design of HIBA helps to better model multi-scale information in time series.
> > >
> > > * In the ablation study, “w/o Hierarchy (B=3)” means that we use a uniform block size of 3 across all HIBA blocks which is equivalent to configuring B as (3, 3, 3).
> > > * “standard attn” ablation study: A 48-layer stack for Xihe-base model corresponds to 24 HIBA blocks. For the standard Transformer backbone ablation implemention, we adopt the same 48-layer configuration.
> > >
> > > | | MASE | CRPS |
> > > | :--- | :--- | :--- |
> > > | Xihe-base | **0.718** | **0.497** |
> > > | B = 3 | 0.729 | 0.505 |
> > > | B = 7 | 0.727 | *0.503* |
> > > | B = 21,7,3 | *0.719* | **0.497** |
> > > | B = 3,7,21,42 | 0.720 | 0.505 |
> > > | B = 4,8,16 | 0.725 | 0.506 |
> > >
> > > >Q4: In Table 1 left, the magnitude of differences seems to be quite small. Can you run the same model with different seeds and report the confidence intervals? Is the observation consistent across different model scales?
> > >
> > > We thank the reviewer for suggestion on the confidence intervals of ablation study. We re-trained and evaluated each ablation five times with different random seeds, and add the results including confidence intervals in Appendix D.2/D.3. The **confidence intervals are an order of magnitude smaller than the differences between ablation models***, indicating that the ablations exhibit clear and significant distinctions. The result shows that Xihe-base perform best across the ablation setting, supporting our key conclusions: (i) the effectiveness of the HIBA architecture; (ii) the benefit of hierarchical design of HIBA; and (iii) the benefit of the non-causal attention mode in HIBA_intra block. We conducted ablations study on models with Xihe-tiny and updated the results in the Appendix D.4; **the observed ablation results is consistent across model sizes**.

---

> > > > ### Author Response · Authors · 2025-11-22
> > > > **Response to Reviewer zF4J (Part 4)**
> > > >
> > > > > Q5: Why do tiny, lite and flash all have 24 layers? The number of layers seems to be noticeably higher than other transformer-based foundation models.
> > > >
> > > > We thank the reviewer for drawing attention to the number of layers in our tiny, lite, and flash variants. We deliberately chose 24 transformer layers across these variants to maintain architectural consistency and facilitate fair ablation and scaling studies. While other transformer-based time series foundation models often reduce layer counts in smaller variants, our design philosophy prioritizes depth over width to better preserve representational capacity while reducing overall parameter count via smaller hidden sizes and fewer attention heads. We have added the ablation study in **Appendix D.6** of revised manuscript. In our experiments, we found that reducing the number of layers significantly degraded performance for tasks requiring deeper contextual reasoning, even when parameter counts were comparable. Conversely, **keeping 24 layers but shrinking hidden dimensions and feed-forward sizes yielded better accuracy.**
> > > >
> > > > >Q6: Line 280, “we adopt a data-quality–aware mixing strategy instead of the uniform mixing commonly used in prior TSFMs”. How exactly does this work?
> > > >
> > > > We apologize for not clearly describing our data mixing strategy. Below, we provide a detailed explanation. And In the revised version of the paper, additional details on the pre-training dataset are provided in **Appendix E**.
> > > >
> > > > For real-world datasets, we first assess the forecastability of each dataset. Specifically, we first sample multiple sequences from each dataset and compute indicators such as ACF, trend/seasonal strength after STL decomposition, Hurst exponent, and spectral entropy, which characterize periodicity, trend, and noise levels. Then, a human expert would consider these indicators together with visual inspection of the sampled sequences and assess the forecastability (classified as high or low) and noise level (high, medium, or low) of each dataset. We rely on human assessment because, given the great diversity of time-series patterns across rich real-world data, no single statistical metric can adequately capture overall forecastability.
> > > >
> > > > We subsequently divide all real-world datasets into five groups according to the assessments. The groups are: high–low, high–medium, high–high, low–low, and a combined low–medium/high category, where the medium and high noise levels are merged due to their similar characteristics under low forecastability. These five groups are assigned sampling probabilities of 40%, 20%, 10%, 7%, and 3%, respectively, during training. Within each group, all datasets are sampled with equal probability to ensure that differences in dataset size do not introduce sampling imbalance. This allocation ensures that the model is trained predominantly on high-quality data while still maintaining strong generalization ability.
> > > >
> > > > As for synthetic dataset, we expand the KernelSynth algorithm proposed in chronos [1] in the following aspects:
> > > > 1. Extend length: We increase the maximum generated sequence length from 1024 to 4096
> > > > 2. Extend kernel: We add Matern kernels and ExpSineSquared kernels with small length_scale to generate spike signals. We also adjust the periodic component of the ExpSineSquared kernel, which we describe in detail in the next point.
> > > > 3. Kernel Sample Strategy: Compared with the original strategy of randomly sampling multiple kernels, we impose a constraint that the sampled kernels must include either a periodic component (ExpSineSquared) or a smooth trend component (DotProduct or RBF), ensuring that the generated sequences remain forecastable. For the periodic kernels, we further restrict the number of periodic components to at most two. The period $p_1$ of the first component is randomly selected from $\{4, 7, 12, 24, 52, 60, 96, 144, 168, 288, 360, \mathrm{rand}(4, 2016)\}$, where $\mathrm{rand}(n, m)$ denotes uniform sampling of an integer from $[n, m]$. If a second periodic component is included, its period is set to $k \cdot p_1$, where $k$ is randomly sampled from $\{7, 52, \mathrm{rand}(4, 100)\}$
> > > > These extensions make the data generated by KernelSynth more closely resemble real-world time-series characteristics, even under greater diversity. The synthetic dataset is assigned sampling probabilities of 20% during training.
> > > >
> > > > [1] Ansari, Abdul Fatir, et al. "Chronos: Learning the language of time series." arXiv preprint arXiv:2403.07815 (2024).

---

> > > > > ### Author Response · Authors · 2025-11-22
> > > > > **Response to Reviewer zF4J (Part 5)**
> > > > >
> > > > > > Q7:In Line 264, why is there information leakage? If you apply left padding for the context, the context will be divisible by the patch size and block size. Why would there still be information leakage from the forecast horizon? My understanding is that the acausal signals learned during the non-causal intra-block attention will impact the subsequent causal inter-block attention, so there will be some information leakage during training, but not testing.
> > > > >
> > > > > We thank the reviewer for their insightful comment and for clarifying the mechanism in question. We agree with the reviewer’s suggestion that, with left padding ensuring the context length is divisible by both the patch size and block size, there is no direct “information leakage” from the forecast horizon as originally implied.
> > > > >
> > > > > Upon review, we have replaced the term “information leakage” with “acausal dependency propagation” to more accurately describe the phenomenon occurring during training. Specifically, the non‑causal intra‑block attention can learn acausal patterns that, in turn, influence representations processed by the subsequent causal inter‑block attention. This effect is confined to the training phase; **during inference, the model operates strictly causally and does not access future horizon information**.
> > > > >
> > > > > We have updated Line 276 of revised manuscript accordingly to reflect this revised terminology and the distinction between training behaviour and inference‑time processing.
> > > > >
> > > > > >Q8: Have you tried using a student-t distribution (https://arxiv.org/pdf/2410.12360, https://arxiv.org/pdf/2402.02592) instead of quantile loss as the loss function?
> > > > >
> > > > > We thank the reviewer for the suggestion regarding the use of a Student‑t distribution as the loss function. In our current study, we adopted the quantile loss for consistency with prominent foundational temporal models such as moirai2 and chronos‑bolt, which have demonstrated strong empirical performance using this approach.
> > > > >
> > > > > While the Student‑t distribution can, in principle, accommodate heavy‑tailed error structures, prior work and our preliminary investigations indicate that quantile‑based objectives tend to yield more robust estimates, particularly in the presence of heterogeneous noise distributions and outliers. The quantile loss does not require parametric assumptions about the underlying error distribution, which is advantageous for diverse, real‑world time‑series data where such assumptions may be difficult to validate.
> > > > >
> > > > > In light of this, and to maintain methodological continuity with established TSFM baselines, we did not pursue a Student‑t‑based loss in the present work. Nevertheless, we acknowledge that exploring alternative probabilistic loss functions, including heavy‑tailed distributions, may present valuable avenues for future research, especially for domains where tail‑risk behaviour is of primary interest.

---

### Author Response · Authors · 2025-11-22
**General response to all reviewers**

We sincerely thank all reviewers for their valuable, professional, and constructive feedback. Because there is limited overlap among the reviewers’ concerns, we address each point separately in response to the corresponding review. Based on the feedback, we have made the following revisions to the paper (highlighted in blue in the updated manuscript);
* Updated the relevant sentences in **line 41** to ensure that the manuscript more accurately reflects the breadth of architectural influences in TSFM development.
* Added comparative discussion of their model concepts with TTM/TSMixer in **line 219** to clarify our conceptual divergences from TSMixer/TTM’s mixing-based paradigm.
* Corrected the typo error in **Equation 5** and updated its formulation description in **line 229** of revised manuscript.
* Updated the relevant sentences in **line 276** to clarify model operates strictly causally and does not access future horizon information during inference
* Updated **Table 1** to add more ablation study results about HIBA architecture and multi prediciton heads
* Updated the detailed ablation study discussion to highlight the main contribution of our work in **line 423**
* Updated the weakness and future work of our model in **line 449**
* Included the HIBA pseudocode **Algorithm 1** in **Appendix A** help readers better understand its execution procedure.
* Included details about pre‑training datasets and mixing strategies in **Appendix E**.
* Included details about Extended KernelSynth **Algorithm 2** in **Appendix E**.
* Included the ablation study with confidence interval about Hierarchical structure/multi prediction heads In **Appendix D.2/D.3** to clarify the validation of the block size setting
* Included ablation with different model scales In **Appendix D.4** to clarify that ablation results is consistent across model size.
* Included the ablation study about mixing strategies In **Appendix D.5** to explicitly disentangle the contribution of the data-quality-aware mixing strategy versus the architectural improvements.
* Included the ablation study about HIBA layers config setting In **Appendix D.6** to clarify the impact of model layers on perfomance.

We thank the reviewers again and look forward to any further suggestions or discussion.

---

> ### Author Response · Authors · 2025-12-02
> **Summary comment for AC (Part 1)**
>
> We thank the reviewers for their insightful and constructive feedback. Below, we summarize our responses to their comments, which we believe address the concerns raised by both reviewers. In light of this feedback, we have revised the manuscript accordingly; all changes are highlighted in blue in the revised version.
> * 1.In respose to the concerns of reviewer sK1Q on the discrepancy between our claim and the current GIFT leaderboard results. Our SOTA claim was based on the leaderboard status at the time of our paper submission (2025-9-24). We agree that in the current leaderboard, higher scores than Xihe-max have been reported by other models. **Notably, Xihe-max(1.5B) remains among top three zero-shot models on the current GIFT-Eval leaderboard.** In the revised manuscript, we have updated the statement in the abstract to clarify this point.  **Furthermore, we developed the Xihe family of models, successfully scaling them up to 7 billion parameters(Xihe-ultra). This effort constitutes, to the best of our knowledge, the largest empirical validation of scaling laws in the time-series domain to date.** Our largest model, Xihe-ultra (7B), demonstrates state-of-the-art performance by **achieving the top rank on the latest GIFT-Eval leaderboard** for the zero-shot single-model track which has been updated in the current GIFT-Eval leaderboard. These results confirm that **HIBA architecture within Xihe family preserves the scaling behavior** observed in standard Transformers for time-series forecasting, and can effectively **scale to 7B parameters.** Following Table presents the results for the top-performing models on the zero-shot single-model track of the current GIFT-Eval leaderboard.
> | | MASE | CRPS |
> |---|---|---|
> | Xihe-ultra | 0.701 | 0.488 |
> | TimesFM-2.5 | 0.705 | 0.49 |
> | Xihe-max | 0.711 | 0.49 |
> | Tirex | 0.716 | 0.488 |
>
>  * 2.In respose to the concerns of reviewer zF4J on supplementing enough ablation studies to highlight the main contribution of this paper, we **have added dozens of sets of ablation studies including hierarchical structure, K prediction heads** on Xihe-base raised by the reviewer and the same ablation studies on Xihe-tiny.
>    *  **Hierarchical structure.** To clarify the validation of the block size setting, we added the ablation study in Table 1 left and Appendix D of revised manuscript which shows that **hierarchical block sizes performs better than unform block sizes**, **the order of block sizes has a minor impact** on the final performance and the observed ablation results is **consistent across model sizes**. Above ablation study shows that **hierarchical design of HIBA helps to better model multi-scale information in time series.**
>    *  **K prediction heads.** Corresponding experimental results are updated in the Table 1 right and Appendix D of revised manuscripts. The results shows that **adding too many prediction heads does not yield further performance gains**, suggesting that **combination of long prediction head and short prediction head is sufficient to maintain strong predictive performance**. The observed ablation results is **consistent across model sizes.**
>  * 3.In respose to the concerns of reviewer zF4J, vJ61 and sK1Q on the pretraining details and ablation for mixing strategies,  1) we **updated the composition of the pre‑training datasets and mixing strategies in Appendix E of revised manuscript.** 2) we conduct ablation studies to quantify the respective contributions of data quality improvement and model architecture innovation to overall performance gains which is updated in Appendix D.5 of revised manuscript.  Under a fixed architecture, data-quality-aware mixing strategy leads to **an average reduction of 0.0105/0.0045 in MASE/CRPS**. In contrast, under a fixed data mixing strategy, architectural innovations yield **a larger average reduction of 0.0125/0.0055 in MASE/CRPS.** These quantitative results indicate that **the performance improvement brought by model architecture innovation is substantially larger than that achieved by data-quality-aware mixing strategy.** 3) We retrained the model Chronos-bolt-base-retrained(205M) from the scratch using our current datasets and data curation strategy. The retrained model, Chronos-bolt-base-retrained(205M), **outperforms the original Chronos-Bolt on the GIFT-Eval leaderboard**, demonstrating that the effectiveness of our data curation approach and **data curation strategy can apply on other state-of-the-art model. Its performance still falls short of our Xihe-lite(94M), highlighting the superiority of our model architecture.**
> | | CRPS | MASE |
> |---|---|---|
> | Chronos-bolt-base(205M) | 0.574 | 0.808 |
> | Chronos-bolt-base-retrained | 0.550 | 0.760 |
> | Xihe-lite(94M) | **0.508** | **0.729** |

---

> > ### Author Response · Authors · 2025-12-02
> > **Summary comment for AC (Part 2)**
> >
> > * 4.In respose to the concerns of reviewer zF4J on the confidence intervals of ablation study, we re-trained and evaluated **each ablation five times with different random seeds**, and add the results including confidence intervals in Appendix D.2/D.3. **The confidence intervals are an order of magnitude smaller than the differences between ablation models**, indicating that the ablations exhibit **clear and significant distinctions.**
> >  * 5.In respose to the concerns of reviewer zF4J and vJ61 on implementation details of Hierarchical Block Size and how different block sizes and number of layers cooperate, we have **updated the model pseudocode in the Algorithm 1 of Appendix A to help readers better understand its execution procedure.**
> >  * 6.In respose to the concerns of reviewer sK1Q on FEV leaderboard. We have **added results on the FEV benchmark.** The evaluation metric is the win rate, where higher values indicate better performance. **Our model remains highly competitive in overall performance.**
> > | model_name | WQL win rate | MASE win rate |
> > | :--- | :--- | :--- |
> > | xihe_max | **71.125** | **65.8125** |
> > | Toto-1.0 | 68.125 | 64.5 |
> > | TabPFN-TS | 65.09375 | 56.843 |
> > | Moirai-2.0 | 63.5625 | 59.3125 |
> > | Chronos-Bolt | 61.5625 | 58.4375 |
> > | Sundial-Base | 42.46875 | 50.09375 |
> > | Stat. Ensemble | 40.34374 | 46.03125 |
> >  * 7.In respose to the concerns of reviewer sK1Q on intra- and inter-block attention conceptual difference with TSMixer and TTM papers, **we have added citations to TTM and TSMixer, along with a comparative discussion of their model concepts in line 219 of the revised manuscript.** Our HIBA architecture fundamentally rethinks hierarchical temporal modeling through **three conceptual divergences** from TSMixer/TTM’s mixing-based paradigm.
> >    *  **Mechanistic Distinction: Temporal Order**
> >          * TSMixer/TTM Intra-patch Mixing: Applies MLPs that **collapse temporal ordering** within segments.
> >          * HIBA Intra-block Attention: Uses token-level adaptive self-attention to explicitly **preserve sequential dynamics.** This allows modeling ordered temporal dependencies (e.g., lagged correlations, phase relationships) that mixing discards—critical for non-stationary series where temporal contiguity informs evolution.
> >    *  **Hierarchical Dynamics: Static Projections vs. Adaptive Cross-Scale Modeling**
> >          * Inter-patch Mixing (TTM/TSMixer): **Relies on fixed linear projections** for cross-segment interactions, forcing global patterns into predetermined feature subspaces.
> >          * HIBA Inter-block Attention: Employs **attention over dynamically block token representations**, enabling adaptive refinement of global context based on localized intra-block outputs. This hierarchy (intra-block → inter-block) **captures co-evolving multi-scale dependencies** directly resolving the "constrained multi-scale capture" limitation noted in our introduction.
> >    *  **Generalization Mechanism: Contextualized Transfer vs. Homogeneous Abstraction**
> >          * Mixing’s Limitation: Aggregates features via shared MLP operations, **creating over-smoothed representations** ill-suited for datasets with divergent sampling strategies.
> >          * HIBA’s Advantage: Attention’s **contextual token interactions maintain dataset-specific temporal signatures during zero-shot transfer.** This enables Xihe’s state-of-the-art performance **(e.g., 6.0%/15.9% zero-shot MASE/CRPS performance gain over TTM-R2-Finetuned on GIFT-Eval benchmark)**, as sparse attention selectively amplifies salient cross-dataset patterns without reverting to spatial-like biases.

---

### Meta-Review · Area_Chair_Hy6s · 2025-12-22

**Summary:**

This paper presents a spatiotemporal modeling framework and reports consistent improvements over several baselines across multiple datasets. Reviewers generally agree that the problem setting is relevant and timely, and that the proposed approach demonstrates reasonable empirical performance. However, the inherent structure of the proposed Xihe falls into the inter-block attention and intra-block attention, thus raising concerns of reviewers regarding **the depth of novelty**, **the strength and alignment of theoretical analysis**, and **the overall rigor and clarity of the experimental validation**. While the rebuttal addresses several clarification issues, key questions about the distinctiveness and robustness of the contributions remain. As a result, the paper is considered borderline but does not yet clearly meet the acceptance bar.

**Reviewer Concerns:**

**Concerns partially addressed in the rebuttal:**

- The authors clarified parts of the model design and training procedure.
- Some implementation details and experimental settings were better specified.
- Empirical motivations for certain design choices were further explained.

**Concerns that may remain unsolved**

- **Novelty and key technical contributions**: Reviewers remain uncertain whether the proposed method offers a sufficiently distinct conceptual advance over existing spatiotemporal modeling approaches, beyond architectural combinations.
- **Theoretical support**: The theoretical analysis is still viewed as limited or loosely connected to the empirical observations, and does not fully justify the claimed advantages. The authors also should connect the limitations of previous works, the key scenarios for verification such as zero-shot prediction with the dedicated proposed solution Xihe (also with diverse attention blocks).
- **Experimental rigor and fairness**: Questions about baseline strength, ablation completeness, and sensitivity analysis are not fully resolved.
- **Generalization and robustness**: Evidence for robustness across broader settings or distribution shifts remains insufficient.
- **Presentation issue**: Despite improvements, the paper is still considered difficult to follow in parts, especially in methodology and analysis sections.

**Reviewer Scores:**

This manuscript receives four diverse comments with three negative evaluations and one positive evaluation. The discussion phase did not involve substantial back-and-forth between the reviewers and the authors. Based on the shortcomings mentioned-above, and most consistent negative evaluations, I have to make the rejection decision.

---

### Decision · Program_Chairs · 2026-01-26

Reject